# Fatty acid metabolic reprogramming via mTOR-mediated inductions of PPARγ directs early activation of T cells

Mulki Angela[1,*], Yusuke Endo[1,*], Hikari K. Asou[1], Takeshi Yamamoto[1], Damon J. Tumes[1,2], Hirotake Tokuyama[3], Koutaro Yokote[3] & Toshinori Nakayama[1,4]

To fulfil the bioenergetic requirements for increased cell size and clonal expansion, activated T cells reprogramme their metabolic signatures from energetically quiescent to activated. However, the molecular mechanisms and essential components controlling metabolic reprogramming in T cells are not well understood. Here, we show that the mTORC1–PPARγ pathway is crucial for the fatty acid uptake programme in activated CD4[+] T cells. This pathway is required for full activation and rapid proliferation of naive and memory CD4[+] T cells. PPARγ directly binds and induces genes associated with fatty acid uptake in CD4[+] T cells in both mice and humans. The PPARγ-dependent fatty acid uptake programme is critical for metabolic reprogramming. Thus, we provide important mechanistic insights into the metabolic reprogramming mechanisms that govern the expression of key enzymes, fatty acid metabolism and the acquisition of an activated phenotype during CD4[+] T cell activation.

[1] Department of Immunology, Graduate School of Medicine, Chiba University, 1-8-1 Inohana Chuo-ku, Chiba 260-8670, Japan. [2] South Australian Health and Medical Research Institute, North Terrace, Adelaide South Australia 5000, Australia. [3] Department of Clinical Cell Biology and Medicine, Graduate School of Medicine, Chiba University, 1-8-1 Inohana Chuo-ku, Chiba 260-8670, Japan. [4] AMED-CREST, AMED, 1-8-1 Inohana Chuo-ku, Chiba 260-8670, Japan. * These authors contributed equally to this work. Correspondence and requests for materials should be addressed to T.N. (email: tnakayama@faculty.chiba-u.jp).

After antigenic stimulation through the T-cell receptor (TCR), quiescent naive T cells undergo clonal expansion and initiate immune responses to pathogens[1]. TCR-mediated signal transduction is crucial for T-cell activation, proliferation and efficient differentiation into effector cells[1,2]. Especially, T-cell co-stimulation via CD28 and TCR engagement drives rapid proliferation through the activation of PI3K/Akt and mammalian target of rapamycin (mTOR) signalling pathways[3,4]. mTOR integrates signalling pathways associated with nutrient levels, energy status, cell stress responses and TCR-mediated and growth factor-mediated signalling, and can induce multiple outcomes including cell growth, proliferation and changes in metabolic programmes[5,6]. To fulfil the energetic requirements associated with activation and rapid proliferation, T cells switch their metabolic programme from fatty acid β-oxidation and catabolic metabolism to aerobic glycolysis and anabolic metabolism[7]. Naive T cells are metabolically quiescent and produce ATP by breaking down glucose, fatty acids and amino acids to fuel oxidative phosphorylation[8]. By contrast, activated effector T cells switch to a high dependency on aerobic glycolysis and amino acid transport to supply ATP and NADH molecules required to sustain energetic metabolism and mitochondrial-membrane potential[9–11]. Conversely, inappropriate nutrient uptake or metabolic inhibition prevents T-cell activation and rapid proliferation[12]. If prolonged, this metabolic inhibition can lead to T-cell anergy[13] or apoptosis.

Antigenic stimulation-dependent metabolic reprogramming is accomplished by dynamic changes in the expression of metabolic enzymes downstream of mTOR activation and the induction of transcription factors such as Myc, Hif1a and Srebp1/2 (refs 14,15). CD28-mediated activation of the PI3K pathway is necessary for the induction of glucose uptake via surface expression of the GLUT1 glucose transporter[10,16]. The metabolic transition towards increased aerobic glycolysis and anabolic pathways in activated T cells is reminiscent of metabolic profiles in tumour cells and may represent a general metabolic reprogramming during rapid T-cell activation and proliferation[17,18]. The transcription factor Myc has an essential role in the induction of aerobic glycolysis and glutaminolysis by regulating enzyme expression in activated T cells[19]. Hif1α, which is induced by hypoxia and also by antigen stimulation or inflammatory cytokines, promotes glycolysis in differentiating T helper 17 (Th17) cells and enhances Th17 cell differentiation[20,21]. Both Hif1α stabilization in conditions of normoxia and sustained upregulation of Myc are dependent on mTORC1 activation after antigenic stimulation[22]. Another important component in the metabolic reprogramming of activated T cells is increased lipid biosynthesis. In activated CD8+ T cells, sterol regulatory element-binding proteins (SREBPs) are required to meet the lipid demands that support effector responses[23]. The maturation of SREBPs in CD8+ T cells is sensitive to rapamycin during T-cell activation. Thus, the metabolic checkpoint imposed by TCR-mTOR signal axis has an instructive role in integrating immunological and metabolic input to direct T-cell function.

The nuclear receptor peroxisome proliferator-activated receptor gamma (PPARγ) is known as a regulator of adipocyte differentiation[24,25]. PPARγ has a critical role in lipid metabolism, promoting free fatty acid uptake and triacylglycerol accumulation in adipose tissue and liver[24]. In addition to the well-studied effects of PPARγ on metabolic systems, several pieces of evidence suggest that PPARγ is also an important regulator of cells of the immune system including T cells[26]. Reports suggest that PPARγ negatively influences the differentiation of Th17 cells[27,28]. Other groups showed a critical role for PPARγ in naturally occurring regulatory T cells (nTreg) and adipose tissue resident Treg cell function[29]. Despite the many anti-inflammatory effects of PPARγ, Pparg deficient CD4+ T cells lack the ability to induce systemic autoimmunity following adoptive transfer into a lymphopenic host[30]. Therefore, the overall biological significance of PPARγ in T-cell function is controversial, and the role of PPARγ in the regulation of fatty acid metabolism in CD4+ T cells is unknown.

The transcriptional regulation of fatty acid uptake and de novo fatty acid synthesis, and the relative contribution of each pathway to the activation of CD4+ T cells is unclear. Here, we demonstrate that the signalling axis of TCR–mTORC1–PPARγ and TCR–mTORC1–SREBP1 is essential for the activation of the fatty acid metabolic programme in activated CD4+ T cells. Chromatin immunoprecipitation (ChIP) analysis reveals an important role for PPARγ in the induction of genes associated with fatty acid uptake. Moreover, both gene silencing and pharmacological inhibition of PPARγ results in attenuated cell activation, clonal expansion, and metabolic reprogramming during antigenic stimulation both in vitro and in vivo. Extrinsic fatty acid supplementation restores activation, rapid proliferation and survival under fatty acid-free conditions. Thus, the PPARγ-dependent fatty acid uptake programme is crucial for metabolic reprogramming in naive and memory CD4+ T cells. Our study provides important insights into the fatty acid metabolism processes that permit the rapid proliferation and activation of naive and memory CD4+ T cells after antigenic stimulation.

## Results

**Fatty acid metabolism controls full activation of CD4+ T cells.** Fatty acids (FAs) are known to be essential metabolites for maintaining cell activation, proliferation and function in rapidly proliferating cells such as tumour cells. To elucidate the role of fatty acids in the activation of CD4+ T cells, we first analysed the amounts of lipid metabolites in naive and stimulated CD4+ T cells using a mass-spectrum-based metabolomics approach. As shown in Fig. 1a and Supplementary Fig. 1a, activated CD4+ T cells 24 and 48 h after in vitro stimulation with anti-TCR mAb and anti-CD28 mAb expressed markedly higher amounts of lipid metabolites, including fatty acids, compared with naive CD4+ T cells (time 0). In contrast, the amounts of carnitines, which are required for fatty acid β-oxidation were decreased after activation, as reported previously[19]. We next assessed the expression of genes associated with fatty acid biosynthesis, fatty acid uptake and lipolysis programmes in naive and activated CD4+ T cells (Fig. 1b). Quantitative real-time PCR analysis detected upregulation of mRNAs encoding the enzymes involved in fatty acid biosynthesis including Acaca, Elovl1, Fads2, Scd1, Scd2, Acsl3 and Fasn (Fig. 1b). Similarly, the expression of genes encoding the enzymes in fatty acid uptake programme such as Ldlr, Lrp8, Scarb1 and Vldlr, and lipolysis including Dbi, Fabp5 and Plin2 was increased after antigenic stimulation (Fig. 1c). Interestingly, the expression of Cd36 was not increased but rather decreased after antigenic stimulation. The expression of surface CD36 was also unchanged or slightly decreased in activated cells as compared with naive CD4+ T cells in parallel with mRNA expression (Supplementary Fig. 1b). We next tested whether the activated CD4+ T cells acquired free fatty acids from the external environment using fluorescently labelled palmitate (Bodipy FLC16)[31–33], and as expected, activated CD4+ T cells acquired higher levels of palmitate as compared with naive CD4+ T cells (Fig. 1d). We assessed the kinetics of fatty acid uptake and proliferation after antigenic stimulation. Naive CD4+ T cells gradually acquired extracellular fatty acid from 6 to 24 h after TCR stimulation, and they appeared to start proliferation thereafter (Fig. 1e). We next compared proliferation of activated CD4+ T cells cultured in

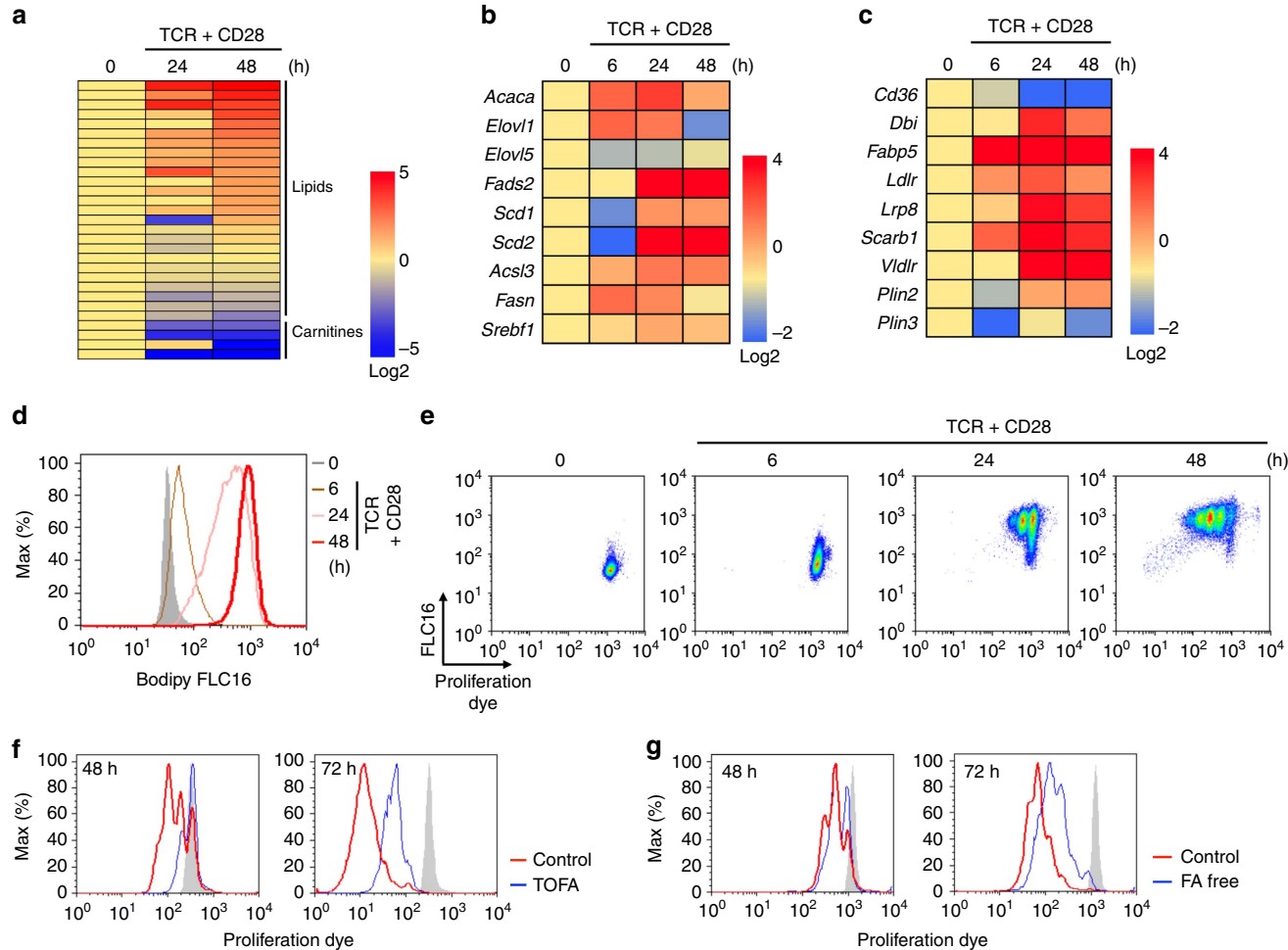

**Figure 1 | FA metabolism is required for the full activation of CD4$^+$ T cells.** (**a**) Metabolome analysis of CD4$^+$ T cells collected at the indicated times after TCR stimulation. The log2 value for each metabolite represents the average of duplicates and the amount of each metabolite in naive CD4$^+$ T cells was set to 0 (see also Supplementary Fig. 1). (**b**) qRT-PCR analyses of the relative expression of the genes encoding the enzymes in fatty acid biosynthesis programme in CD4$^+$ T cells collected at the indicated times after TCR stimulation. The heat map represents the log2 value of the relative mRNA expression level (see colour scale). (**c**) qRT-PCR analyses of the relative expression of the genes encoding the enzymes and transporter in fatty acid uptake and lipolysis programmes in CD4$^+$ T cells collected at the indicated times after TCR stimulation as in (**b**). (**d**) Representative plots of Bodipy FLC16 in CD4$^+$ T cells collected at the indicated times after TCR stimulation are shown. (**e**) Naive CD4$^+$ T cells were labelled with e670 proliferation dye and stimulated with immobilized anti-TCRβ mAb and anti-CD28 mAb in the presence of Bodipy FLC16. Representative profiles of e670 and Bodipy FLC16 in CD4 T cells collected at the indicated times after TCR stimulation are shown. (**f**,**g**) Naive CD4$^+$ T cells were labelled with e670 proliferation dye and stimulated with immobilized anti-TCRβ mAb and anti-CD28 mAb in the presence of TOFA (**f**) or under fatty acid-free conditions (**g**). Three technical replicates were performed for qRT-PCR (**b**,**c**). Three independent experiments were performed with similar results (**b**–**g**).

the presence of 5-(tetradecyloxy)-2-furoic acid (TOFA), a pharmacological inhibitor of ACC1 (a master enzyme of fatty acid biosynthesis), or cultured under fatty acid-free conditions. The proliferation of activated CD4$^+$ T cells was substantially inhibited by TOFA treatment (Fig. 1f). We also detected a substantial impairment of proliferation of activated CD4$^+$ T cells during culture under fatty acid-free conditions (Fig. 1g). These results indicate that both fatty acid biosynthesis and fatty acid uptake programmes are required for the early activation and proliferation of CD4$^+$ T cells.

**TCR–mTORC1 signalling axis induces expression of PPARγ.** CD28 co-stimulatory signals synergize with the TCR to promote clonal expansion of activated T cells[3]. Increased fatty acid uptake was observed in activated naive CD4$^+$ T cells in the presence of CD28 co-stimulatory signals (Supplementary Fig. 2a). Lower expression of the genes associated with fatty acid biosynthesis

including *Acaca*, *Elovl1*, *Fads2*, *Scd1*, *Scd2*, *Acsl3* and *Fasn* was detected in activated CD4$^+$ T cells in the absence of CD28 co-stimulation (Supplementary Fig. 2b). We also detected decreased induction of the genes associated with fatty acid uptake such as *Ldlr*, *Lrp8*, *Scarb1* and *Vldlr*, and lipolysis including *Dbi*, *Fabp5* and *Plin2* in activated CD4$^+$ T cells in the absence of CD28 co-stimulation (Supplementary Fig. 2c). Decreased fatty acid uptake and proliferation was observed in the absence of CD28 co-stimulatory signals even under very strong TCR stimulation conditions. Therefore, CD28 may provide a necessary additive signal for fatty acid uptake and rapid proliferation during antigenic stimulation (Supplementary Fig. 2d,e). We next examined the phosphorylation of S6 ribosomal protein, a downstream molecule of the mTOR pathway, in conjunction with the levels of fatty acid uptake (Fig. 2a). Phosphorylation of S6 was positively associated with fatty acid uptake in activated CD4$^+$ T cells. In addition, the expression of genes associated with fatty acid biosynthesis

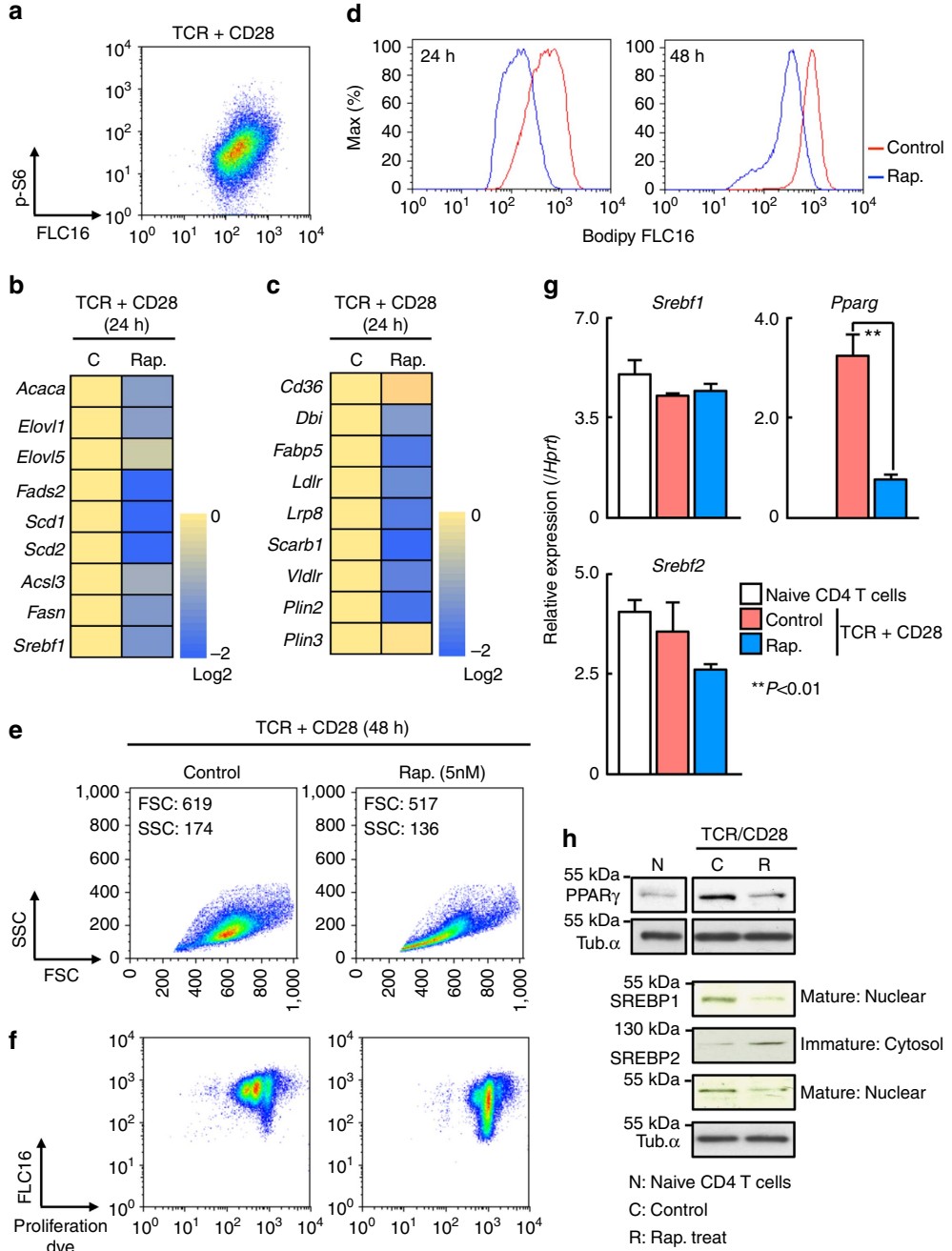

**Figure 2 | TCR/CD28-mTORC1 signal axis induces PPARγ and SREBP1 activation.** (**a**) Intracellular staining profiles of p-S6 protein and Bodipy FLC16 in stimulated CD4$^+$ T cells for 24 h. (**b**) qRT-PCR analyses of the relative expression of the genes encoding the enzymes in fatty acid biosynthesis programme in stimulated CD4$^+$ T cells in the presence of rapamycin. The heat map represents the log2 value of the relative mRNA expression level (see colour scale). The log2 value of each gene in control cells was set to 0. (**c**) qRT-PCR analyses of the relative expression of the genes encoding the enzymes and transporter in fatty acid uptake and lipolysis programmes in stimulated CD4$^+$ T cells in the presence of rapamycin as in **b**. (**d**) Representative plots of Bodipy FLC16 in CD4$^+$ T cells collected at the indicated times after TCR stimulation in the presence of rapamycin are shown. (**e**) Forward and side scatter of live cells 48 h after TCR stimulation in the presence or absence of rapamycin. (**f**) Representative profiles of e670 and Bodipy FLC16 in stimulated CD4$^+$ T cells in the presence of rapamycin 48 h after TCR stimulation are shown. (**g**) qRT-PCR analyses of the relative expression of *Srebf1*, *Srebf2* and *Pparg* in naive CD4$^+$ T cells and activated CD4$^+$ T cells in the presence or absence of rapamycin are shown. (**P < 0.01, Mann–Whitney *U* test). (**h**) Protein levels of SREBP1, SREBP2 and PPARγ in naive CD4 T cells and stimulated cells in the presence or absence of rapamycin. Three technical replicates were performed for qRT-PCR (**b**,**c**,**g**). The mean values with s.d. are shown. **P < 0.01 more than three independent experiments were performed with similar results (**a–h**).

including *Acaca*, *Elovl1*, *Fads2*, *Scd1*, *Scd2*, *Acsl3* and *Fasn* was decreased by rapamycin treatment (Fig. 2b). The expression of the genes associated with the fatty acid uptake programme such as *Ldlr*, *Lrp8*, *Scarb1* and *Vldlr*, and lipolysis including *Dbi*, *Fabp5* and *Plin2* were reduced by inhibition of mTOR in activated

CD4$^+$ T cells (Fig. 2c), accompanied by decreased fatty acid uptake (Fig. 2d). Consistent with these results, rapamycin-treated CD4$^+$ T cells showed decreased cell size and reduced dilution of proliferation dye after stimulation (Fig. 2e,f). mTORC1 activates transcription factor SREBP-1 in cancer cells[18] and induces

PPARγ in adipocyte[34]. The expression of these transcription factors was therefore assessed in activated CD4+ T cells in the presence or absence of rapamycin. A significant decrease in *Pparg* mRNA was detected in response to addition of rapamycin, whereas expression of *Srebf1* and *Srebf2* mRNA was not decreased obviously by the addition of rapamycin (Fig. 2g). However, consistent with previous observations using activated whole splenic T cells cultured with rapamycin[23], we observed a substantial decrease in expression of SREBP1 protein in the nucleus and mature-type nuclear SREBP2 protein in rapamycin-treated CD4+ T cells (Fig. 2h). Taken together, these results indicate that mTORC1 controls fatty acid uptake and biosynthesis programmes through the induction of PPARγ and the activation of SREBP1 in activated CD4+ T cells.

**PPARγ controls gene expression in FA uptake programme.** To better understand how PPARγ and SREBP1 are involved in the regulation of fatty acid biosynthesis and fatty acid uptake programmes in CD4+ T cells, we assessed the expression of genes encoding the enzymes in these metabolic pathways after siRNA-mediated knockdown of *Pparg* and *Srebf1*, or after pharmacological inhibition of PPARγ. Reduced expression of *Pparg* resulted in a reduction in expression of fatty acid uptake and lipolysis genes including *Lrp8*, *Fabp5*, *Ldlr* and *Scarb1* (Fig. 3a). Similar results were obtained from the experiments using the PPARγ inhibitor, GW9662 (Fig. 3b). Interestingly, the expression of CD36, receptor for lipoproteins and long-chain fatty acids, was increased by the inhibition or silencing of *Pparg* (Fig. 3a,b). Similarly, reduced expression of *Srebf1*, but not *Srebf2,* resulted in a reduction of fatty acid synthesis genes including *Scd1*, *Scd2*, *Fads2* and *Fasn* (Fig. 3c). Epigenetic chromatin modifications can regulate selective expression of genes that function in the immune systems[35]. To address the underlying molecular mechanisms by which PPARγ and SREBP1 control fatty acid metabolism, we first examined chromatin status of the genes encoding fatty acid metabolism-associated enzymes in activated CD4+ T cells. We performed ChIP assays with antibodies specific to several histone modifications including the permissive histone mark H3K4-Me3, and the repressive histone mark H3K27-Me3 at the genetic loci associated with fatty acid lipolysis (*Fabp5* locus) and fatty acid biosynthesis programmes (*Scd2* locus) (Supplementary Fig. 3a). At the *Fabp5* locus, naive CD4 T cells showed lower modifications associated with active transcription (H3K4-Me3) and higher modifications associated with genetic repression (H3K27-Me3) than did activated CD4+ T cells (Supplementary Fig. 3b, open and filled bars), suggesting that TCR stimulation induces dynamic chromatin remodelling of these genetic loci. Activated CD4+ T cells cultured in the presence of GW9662 showed reduced modifications of active marks at these gene loci (Supplementary Fig. 3b, red bars). In contrast, the levels of repressive histone marks were increased in GW9662 treated cells (Supplementary Fig. 3b). Likewise, we observed similar pattern of histone modifications at the *Scd2* locus in rapamycin-treated activated CD4+ T cells (Supplementary Fig. 3c,d). Consequently, we next performed ChIP analysis of the genes associated with fatty acid synthesis and uptake using specific antibodies for PPARγ and SREBP1. The binding of PPARγ to the promoters of genes containing PPARγ binding motifs *Fabp5*, *Ldlr*, *Plin2*, *Scarb1* and *Vldlr* was reduced by the treatment of activated CD4+ T cells with GW9662, whereas the binding of PPARγ to *Hprt*p was unaffected by GW9662 treatment (Fig. 3d). Furthermore, the binding of SREBP1 to the promoter region of SREBP target

gene loci such as *Acaca*, *Fads2* and *Scd2* was reduced by the treatment of activated CD4+ T cells with rapamycin (Fig. 3e). These results indicate that PPARγ and SREBP1 directly bind and control the expression of the genes involved in the fatty acid uptake and synthesis programmes.

**PPARγ is essential for the rapid proliferation of CD4+ T cells.** To investigate the role of PPARγ in fatty acid uptake programme more directly, we next examined the effect of PPARγ inhibition on fatty acid uptake and proliferation after antigenic stimulation. As we expected, fatty acid uptake and proliferation after antigenic stimulation was impaired by GW9662 treatment or si*Pparg* transduction (Fig. 4a–c). We also assessed low-density lipoprotein (LDL) or nonpolar neutral lipids uptake using LDL complex or neutral lipids conjugated with fluorescent dye (Supplementary Fig. 4a). Although the uptake of LDL or neutral lipid was induced by TCR stimulation, much less effect of GW9662 on the reduction of the lipid metabolite uptake than fatty acid uptake was detected. We next assessed the role of fatty acid biosynthesis and fatty acid uptake in differentiating Th1, Th2 and Th17 cells. Similar to Th0 cells, the induction of genes associated with fatty acid metabolism was detected in these Th subsets. Consistent with these results, the proliferation of Th1 and Th2 cells was substantially inhibited by TOFA treatment (Supplementary Fig. 4b). Interestingly, the proliferation of Th17 cells was not affected by TOFA treatment. Fatty acid uptake (FLC16 intensity) and proliferation after antigenic stimulation were impaired by GW9662 treatment in these Th subsets (Supplementary Fig. 4c). To further investigate the metabolic state of GW9662-treated CD4+ T cells, we analysed the cellular bioenergetics of activated CD4+ T cells. We found that the extracellular acidification rate (ECAR), a consequence of lactic acid production (which is a marker of glycolysis) and oxygen-consumption rate (OCR), an indicator of oxidative phosphorylation, were lower in GW9662-treated CD4+ T cells than that in control CD4+ T cells in the basal state (Fig. 4d). We next directly assessed mitochondrial function and the rate of acid efflux in GW9662-treated CD4+ T cells by monitoring the OCR and ECAR in response to sequential treatment with the ATPase inhibitor Oligomycin, the uncoupling agent carbonyl cyanide-p-trifluoromethoxyphenylhydrazone (FCCP), and the electron-transport-chain inhibitors Rotenone and Antimycin A (Fig. 4e,f). Control CD4+ T cells demonstrated significantly larger mitochondrial spare respiratory capacity (SRC) as compared with GW9662-treated cells (Fig. 4e). The level of ECAR in a maximum state was also significantly decreased in GW9662-treated CD4+ T cells (Fig. 4f). We also examined the effect of *Pparg* knockdown by siRNA in activated CD4+ T cells in seahorse experiments. Similar to the effect of GW9662, the levels of OCR and ECAR were decreased in *Pparg*-knockdowned CD4+ T cells as compared with control CD4+ T cells (Fig. 4g,h). These results indicate that the PPARγ-induced fatty acid uptake programme is required for metabolic reprogramming in activated CD4+ T cells. To directly assess the role of PPARγ in the early activation of CD4+ T cells under more physiological conditions, the fatty acid uptake and proliferation of antigen specific CD4+ T cells were examined *in vivo*. Briefly, naive CD4+ T cells from OVA-specific DO11.10 Tg mice were transferred into normal syngeneic BALB/c mice, and immunized with OVA/Alum (Fig. 4i). Administration of GW9662 resulted in the suppression of proliferation of CD4+ T cells *in vivo* (Fig. 4j,k). The reduced proliferation after GW9662 treatment was associated with inhibition of fatty acid uptake in these cells (Fig. 4j,k). These results indicate that the induction of PPARγ is critical for the antigen-induced rapid proliferation of CD4+ T cells *in vivo*.

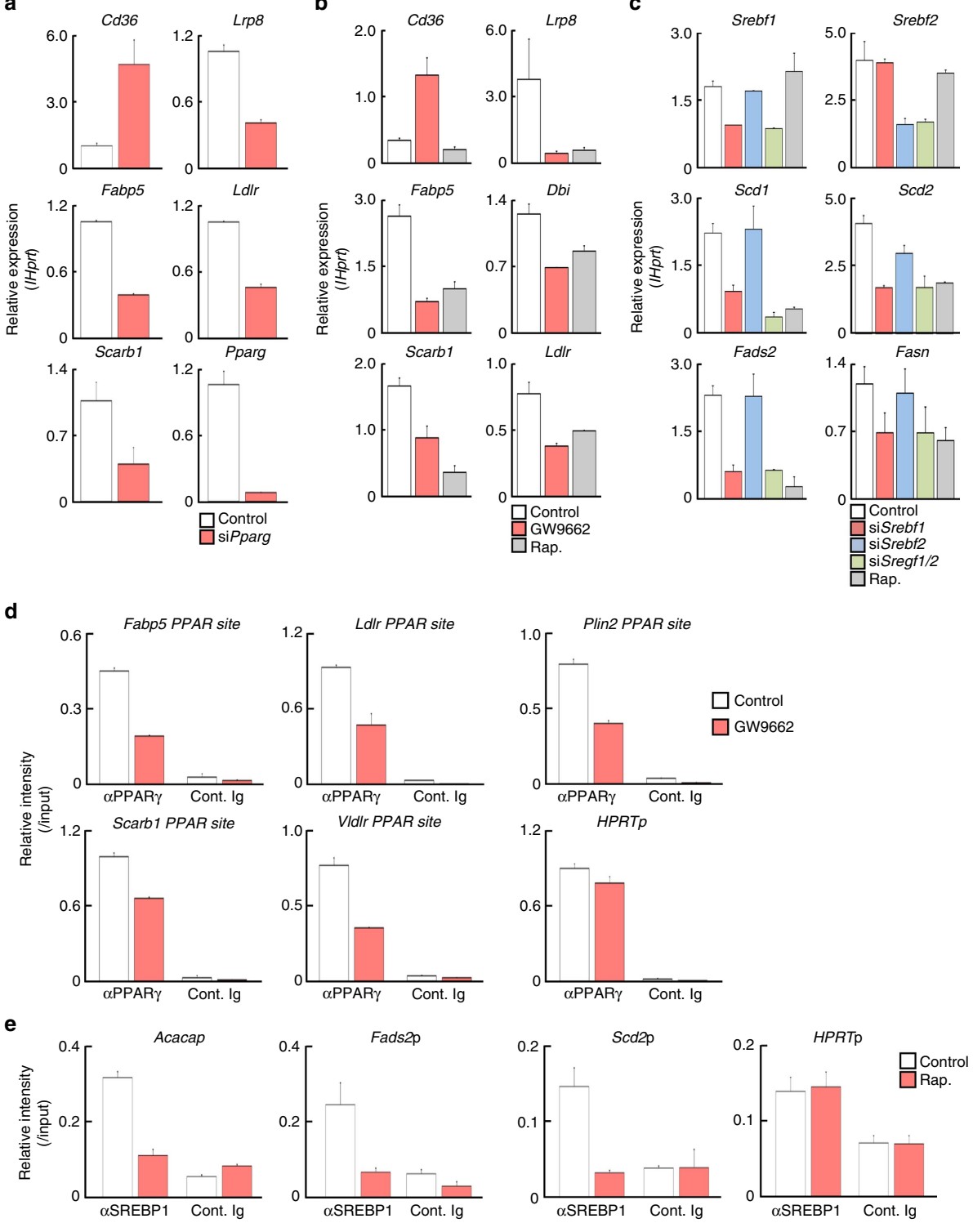

**Figure 3 | PPARγ controls expression of genes associated with FA uptake programme.** (**a**) Effects of silenced *Pparg* on the genes encoding the enzymes and transporters in fatty acid uptake programme in stimulated CD4$^+$ T cells. Naive CD4$^+$ T cells were electroporated with control or *Srebf1* siRNA and cultured for 2 days, and qRT-PCR analyses of the indicated genes are shown. (**b**) qRT-PCR analyses of the indicated genes in activated CD4$^+$ T cells with or without GW9662. (**c**) Effects of silenced *Srebf1* on the genes encoding the enzymes in fatty acid biosynthesis programme in stimulated CD4$^+$ T cells as in **a**. (**d**) ChIP assays were performed with anti-PPARγ at the promoter region of the target gene loci such as *Fabp5*, *Ldlr*, *Plin2*, *Scarb1*, *Vldlr* and *Hprt* from activated CD4$^+$ T cells treated with DMSO (Control) and GW9662 (10 μM). (**e**) ChIP assays were performed with anti-SREBP1 at the promoter regions of *Acaca*, *Fads2*, *Scd2* and *Hprt* from activated CD4$^+$ T cells treated with DMSO (Control) and rapamycin (Rap.; 5 nM). Three technical replicates were performed for qRT-PCR (**a–c**) and ChIP qRT-PCR (**d,e**). Three independent experiments were performed with similar results (**a–e**).

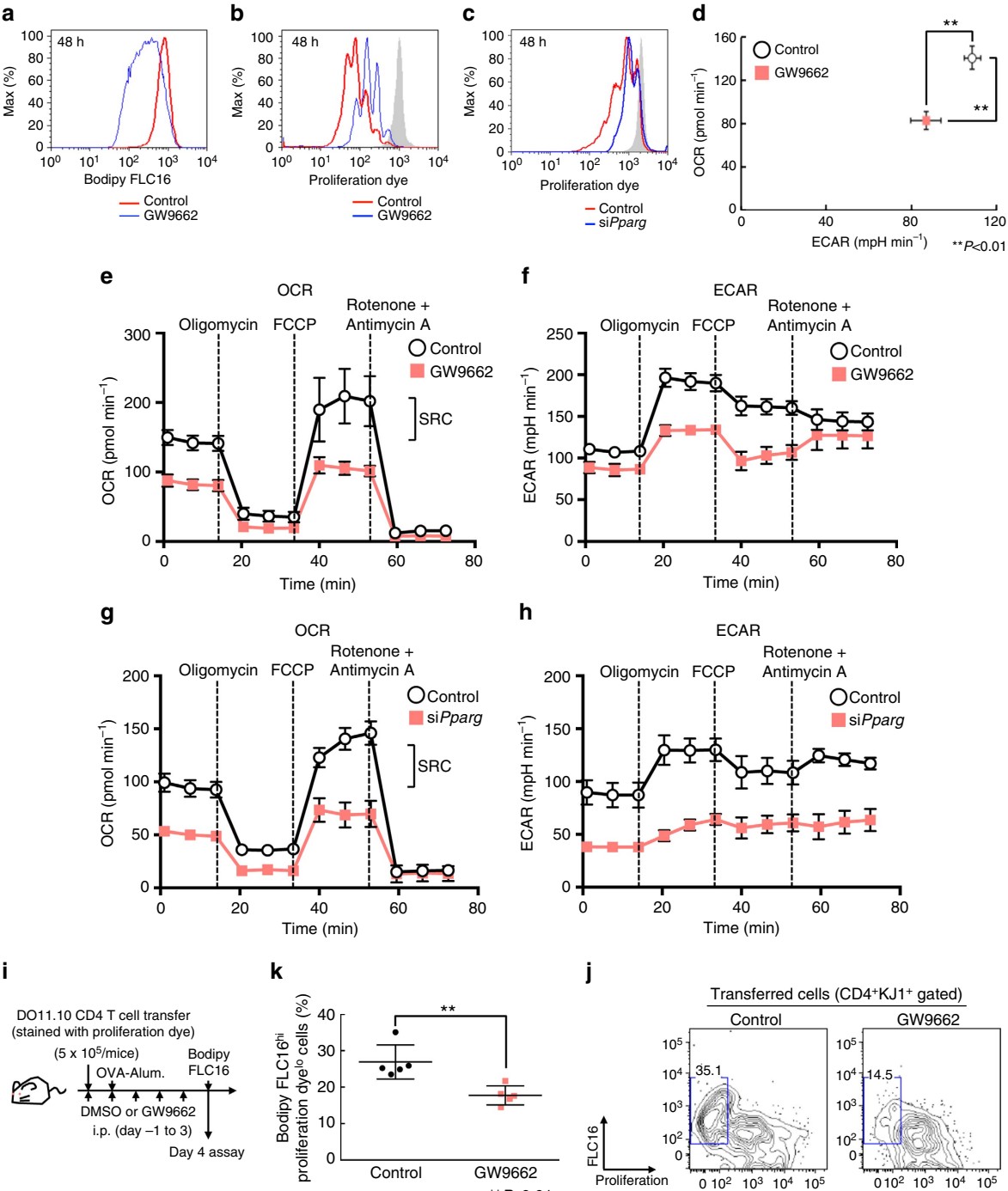

**Figure 4 | PPARγ-dependent FA uptake is essential for rapid proliferation of CD4$^+$ T cells.** (**a**) Representative plots of Bodipy FLC16 in activated CD4$^+$ T cells 48 h after TCR stimulation with or without GW9662 are shown. (**b**) Representative profiles of e670 in stimulated CD4$^+$ T cells in the presence or absence of GW9662. (**c**) Representative profiles of e670 in stimulated CD4$^+$ T cells with or without transduction of siRNA for *Pparg*. (**d**) Basal levels of OCR and ECAR of activated CD4$^+$ T cells for 48 h with TCR stimulation in the presence or absence of GW9662. (**P $< 0.01$, Mann–Whitney $U$ test, $n = 6$ per group) (**e**) OCR of activated CD4$^+$ T cells 48 h after TCR stimulation with or without GW9662 under basal conditions (time point 0) and in response to sequential treatment with Oligomycin, FCCP and Rotenone-Antimycin A. (**f**) ECAR of activated CD4$^+$ T cells 48 h after TCR stimulation with or without GW9662 under basal conditions and in response to sequential treatment with Oligomycin, FCCP and Rotenone-Antimycin A as in **e**. (**g**) OCR of activated CD4$^+$ T cells 48 h after TCR stimulation with or without transduction of siRNA for *Pparg* as in **e**. (**h**) ECAR of activated CD4$^+$ T cells 48 h after TCR stimulation with or without transduction of siRNA for *Pparg* under basal conditions and in response to sequential treatment with Oligomycin, FCCP and Rotenone-Antimycin A as in **f**. (**i**) Experimental protocols for OVA/Alum-induced CD4$^+$ T cell proliferation with the administration of GW9662. (**j**) Representative profiles of e670 and Bodipy FLC16 in activated DO11.10 TCR Tg CD4 T cells collected from control or GW9662 treated OVA-immunized mice are shown. (**k**) The summary data for the profiles of e670 and Bodipy FLC16 in activated DO11.10 Tg CD4$^+$ T cells are shown as frequencies with standard deviations (**P $< 0.01$, Mann–Whitney $U$ test, $n = 5$ per group). Six technical replicates were performed for Seahorse assay (**d–h**). The mean values with s.d. are shown. **P $< 0.01$ Three independent experiments were performed with similar results (**a–h**). Two independent experiments were performed with similar results (**j,k**).

**FA metabolism regulates memory CD4$^+$ T cell activation.** When secondary immune responses occur, memory T cells can mount faster and stronger immune responses as compared with those in the first response[36]. We next investigated whether anabolic fatty acid metabolism was also involved in the regulation of rapid activation of memory CD4$^+$ T cells after antigenic stimulation using *in vivo* generated antigen-specific memory Th2 cells[37–39]. Expression of mRNAs encoding metabolic enzymes and transporters involved in fatty acid synthesis, uptake and lipolysis programmes was rapidly upregulated by antigenic

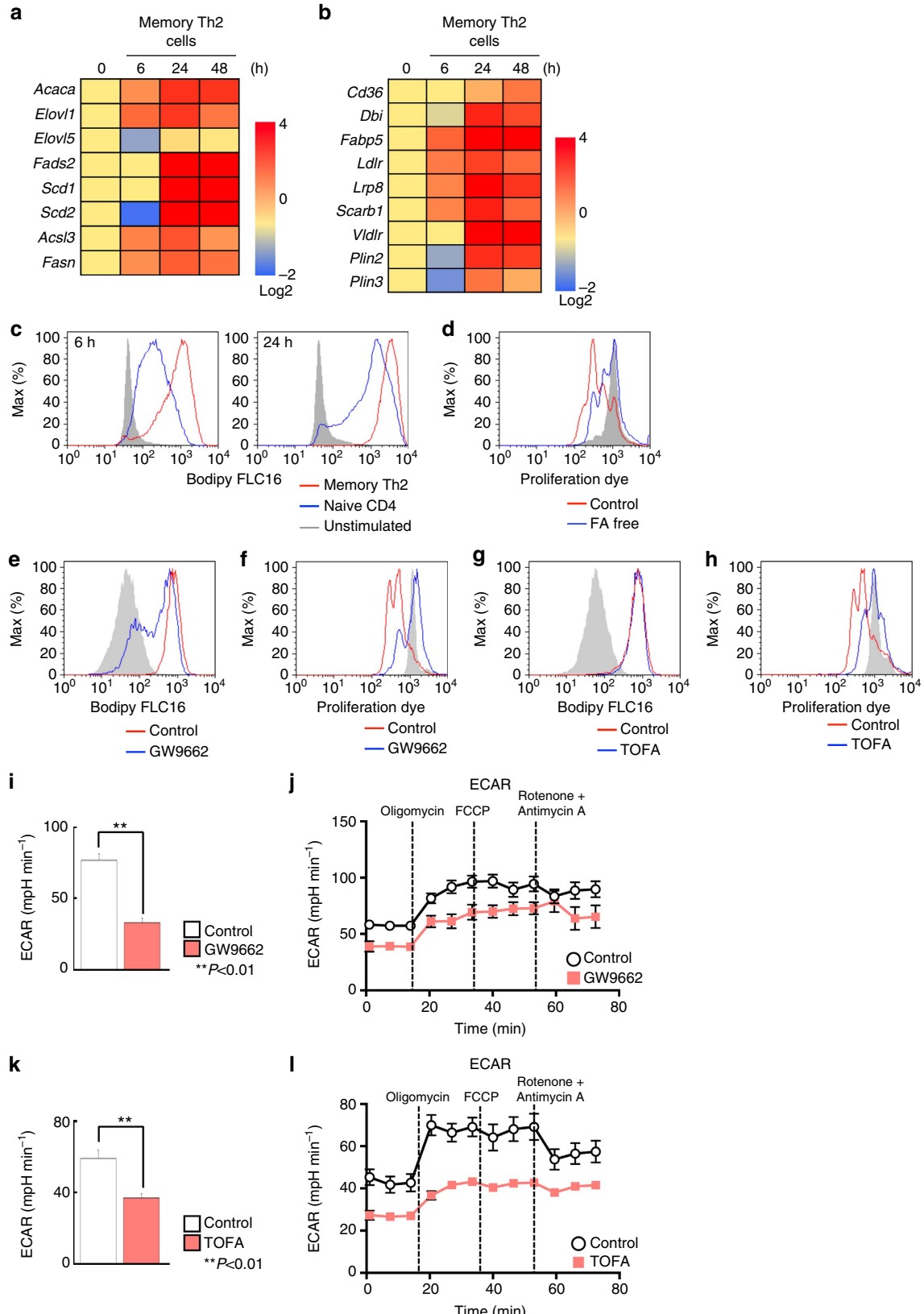

stimulation in memory Th2 cells (Fig. 5a,b). As expected, stimulated memory Th2 cells acquired significantly higher levels of palmitate as compared with unstimulated cells (Fig. 5c). Interestingly, memory Th2 cells acquired extracellular fatty acid more rapidly as compared with naive CD4$^+$ T cells after antigenic stimulation. Similarly, CD44$^{hi}$ memory phenotype CD4 T cells displayed rapid fatty acid uptake consistent with rapid induction of mRNAs encoding the enzymes involved in fatty acid uptake programmes (unpublished data). Consistent with the results of rapid fatty acid uptake, antigenic stimulation-induced memory Th2 cell proliferation was impaired under fatty acid-free conditions (Fig. 5d). In addition, fatty acid uptake and proliferation was suppressed by the treatment of memory Th2 cells with GW9662, as was detected in activated naive CD4 T cells (Fig. 5e,f). Furthermore, TOFA treatment resulted in the inhibition of rapid proliferation without affecting fatty acid uptake in activated memory Th2 cells (Fig. 5g,h). To examine the impact of the inhibition of fatty acid uptake and biosynthesis programmes on the metabolic reprogramming in memory Th2 cells, we next investigated how GW9662 and TOFA influences TCR stimulation-driven changes in cellular metabolism. We measured ECAR following treatment with Oligomycin, and GW9662-treated cells displayed a lower ECAR as was detected in activated naive CD4$^+$ T cells (Fig. 5i,j). Similar results were obtained from TOFA-treated memory Th2 cells (Fig. 5k,l). In contrast to the results of naive CD4$^+$ T cells, OCR in memory Th2 cells was unchanged by the inhibition of these pathways (Supplementary Fig. 5a,b). These results suggest that both fatty acid biosynthesis and fatty acid uptake programmes are essential for the rapid activation and metabolic reprogramming of memory Th2 cells.

**FA supplementation restores activated CD4+ T cell phenotype.** We previously reported that extrinsic fatty acid supplementation restored the function of Acaca$^{-/-}$ Th17 cells[40]. Those data led us to postulate that cellular fatty acid was limiting in TOFA-treated or GW9662-treated CD4$^+$ T cells in the early activation phase. We sought to determine if the supplementation of extracellular fatty acid in cultures could restore the activation of stimulated CD4$^+$ T cells under fatty acid limited conditions. Under fatty acid-free conditions, supplementation of TOFA-treated CD4$^+$ T cells with oleic acid (OA) restored the cellular enlargement and cell survival after antigenic stimulation (Fig. 6a,b). Antigenic stimulation-induced proliferation was restored by the addition of oleic acid to activated CD4$^+$ T cells under fatty acid-free conditions (Fig. 6c). Similar results were obtained in memory Th2 cells (Fig. 6d–f). We checked the effect of a series of fatty acids including saturated fatty acids, long chain saturated fatty acids and polyunsaturated fatty acids (Fig. 6g). While Dodecanoic acid and Lignoceric acid did not

rescue the proliferation of activated CD4$^+$ T cells under fatty acid-free conditions, many of the saturated fatty acids restored the proliferation phenotype of activated CD4$^+$ T cells (Fig. 6g, upper). Especially, supplementation of activated CD4$^+$ T cells under fatty acid-free conditions with Decanoic acid and Myristic acid almost completely restored proliferation. In contrast, the majority of unsaturated fatty acids including polyunsaturated fatty acids other than Oleic acid, Elaidic acid and Alachidonic acid did not rescue the proliferation of activated CD4$^+$ T cells under fatty acid-free conditions (Fig. 6g, lower). Furthermore, we tested the effect of cholesterol in activated CD4$^+$ T cells under fatty acid-free conditions, while extrinsic supplementation of cholesterol did not restore the proliferation or cell survival of activated CD4$^+$ T cells (Supplementary Fig. 6a,b). Regarding the role of intracellular fatty acid on mitochondrial function and fatty acid oxidation, we first analysed the expression of Cpt1, which mediates the first step in long-chain fatty acid import into mitochondria. The expression of Cpt1a and Cpt1b in activated CD4$^+$ T cells was unchanged by TOFA or GW9662 treatment (Supplementary Fig. 6c). The expression of Cpt1c was slightly decreased by the treatment of activated CD4$^+$ T cells with TOFA or GW9662. We determined the feasibility of testing the mitochondrial membrane potential in activated CD4$^+$ T cells with TOFA or GW9662 treatment using mitotracker (Supplementary Fig. 6d). The levels of mitotracker were not substantially affected by TOFA or GW9662 treatment. Furthermore, we also endeavoured to examine the effect of fatty acid supplementation to TOFA-treated activated CD4$^+$ T cells on the mitochondrial respiratory capacity (Supplementary Fig. 6e,f). Extrinsic supplementation of TOFA-treated activated CD4$^+$ T cells under fatty acid-free conditions with Oleic acid significantly restored the levels of SRC and ECAR (Supplementary Fig. 6e,f). These results obtained from mitochondrial experiments suggest that extrinsically supplemented fatty acids could be providing the oxidative substrate in mitochondria. Taken together, these results indicate that fatty acid substitution restored the defects in cellular enlargement, proliferation, survival and metabolic reprogramming of TOFA-treated activated CD4$^+$ T cells.

**FA metabolism is essential for activation of human CD4$^+$ T cells.** Finally, we investigated the role of fatty acid metabolism in human CD4$^+$ T cell activation. Consistent with the results in mouse experiments, rapamycin-treated human CD4$^+$ T cells proliferated only modestly in response to stimulation with anti-CD3 mAb and anti-CD28 mAb (Supplementary Fig. 7a). A decreased induction of genes associated with fatty acid biosynthesis, fatty acid uptake and lipolysis programmes were detected in rapamycin-treated human CD4$^+$ T cells (Supplementary Fig. 7b,c). In addition, a lower

**Figure 5 | FA metabolism is critical for activation and rapid proliferation of memory CD4$^+$ T cells.** (**a**) qRT-PCR analyses of the relative expression of the genes encoding fatty acid biosynthesis enzymes in memory Th2 cells at the indicated times after TCR stimulation. The heat map represents the log2 value of the relative mRNA expression level (see colour scale). (**b**) qRT-PCR analyses of expression of genes encoding the enzymes and transporter in fatty acid uptake and lipolysis programmes in memory Th2 cells collected at the indicated times after TCR stimulation as in **a**. (**c**) Representative plots of Bodipy FLC16 in naive CD4$^+$ T cells and memory Th2 cells collected at the indicated times after antigenic stimulation are shown. (**d**) Representative plot of e670 proliferation dye in memory Th2 cells after antigenic stimulation under fatty acid-free conditions. (**e–h**) Representative plots of Bodipy FLC16 and e670 proliferation dye in stimulated memory Th2 cells in the presence of GW9662 (10 μM) (**e,f**) or TOFA (10 μM) (**g,h**). (**i**) ECAR of memory Th2 cells 48 h after TCR stimulation with or without GW9662 under basal conditions (Time point 0). (**P < 0.01, Mann–Whitney U test, n = 6 per group). (**j**) ECAR of memory Th2 cells 48 h after TCR stimulation with or without GW9662 under basal conditions and in response to sequential treatment with Oligomycin, FCCP and Rotenone-Antimycin A. (**k**) ECAR of memory Th2 cells 48 h after TCR stimulation with or without TOFA under basal conditions. (**P < 0.01, Mann–Whitney U test, n = 6 per group) (**l**) ECAR of memory Th2 cells 48 h after TCR stimulation with or without TOFA under basal conditions and in response to sequential treatment with Oligomycin, FCCP and Rotenone-Antimycin A as in **j**. Three technical replicates were performed for qRT-PCR (**a,b**). Six technical replicates were performed for Seahorse assay (**i–l**). The mean values with s.d. are shown (**i,k**). **P < 0.01 Three independent experiments were performed with similar results (**a–l**).

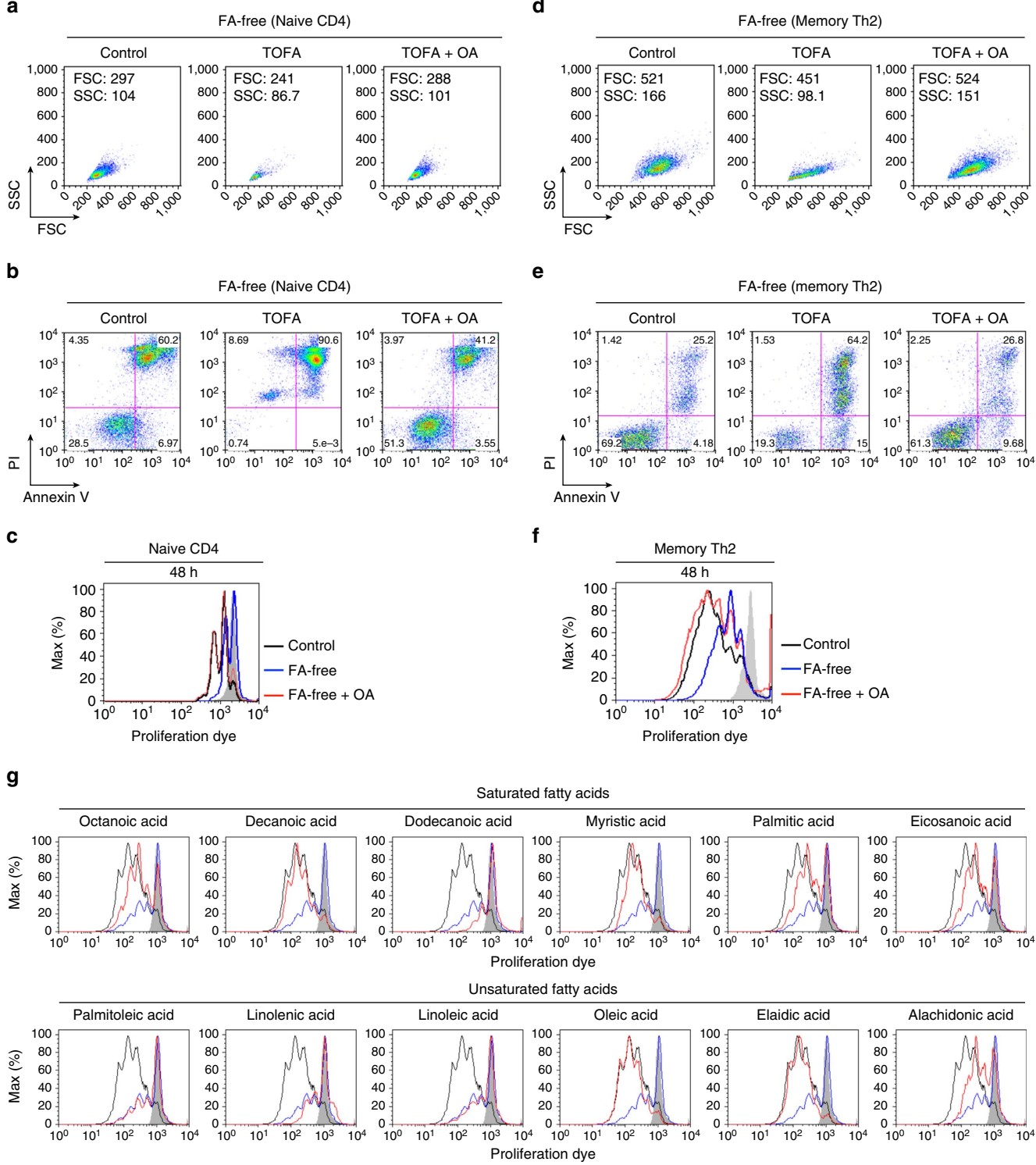

**Figure 6 | Extrinsic FAs restore activated phenotype of CD4$^+$ T cells under FA-free conditions.** (**a**) Naive CD4$^+$ T cells were stimulated with an immobilized anti-TCRβ mAb and anti-CD28 mAb with or without TOFA treatment (10 μM) in the presence or absence of oleic acid (100 μM) under fatty acid-free conditions. Forward and side scatter of live cells after TCR stimulation are shown. (**b**) Susceptibility to apoptosis of stimulated CD4$^+$ T cells was investigated by Annexin V and propidium iodide (PI) staining with similar conditions in the presence or absence of oleic acid (100 μM) under fatty acid-free conditions. (**c**) Representative plot of e670 proliferation dye in activated CD4$^+$ T cells with or without oleic acid treatment under normal or fatty acid-free conditions 48 h after TCR stimulation was shown. (**d–f**) Memory Th2 cells were stimulated with an immobilized anti-TCRβ mAb and anti-CD28 mAb with or without TOFA treatment (10 μM) in the presence or absence of oleic acid (100 μM) under fatty acid-free conditions. Representative profiles of forward scatter and side scatter (**d**), cell survival (**e**) and e670 proliferation dye (**f**) in stimulated memory Th2 cells cultured under the indicated conditions are shown. (**g**) Representative plot of e670 proliferation dye in activated CD4$^+$ T cells with or without a series of fatty acid treatment under normal or fatty acid-free conditions 48 h after TCR stimulation was shown. Three independent experiments were performed with similar results (**a–c**). Two independent experiments were performed with similar results (**d–g**).

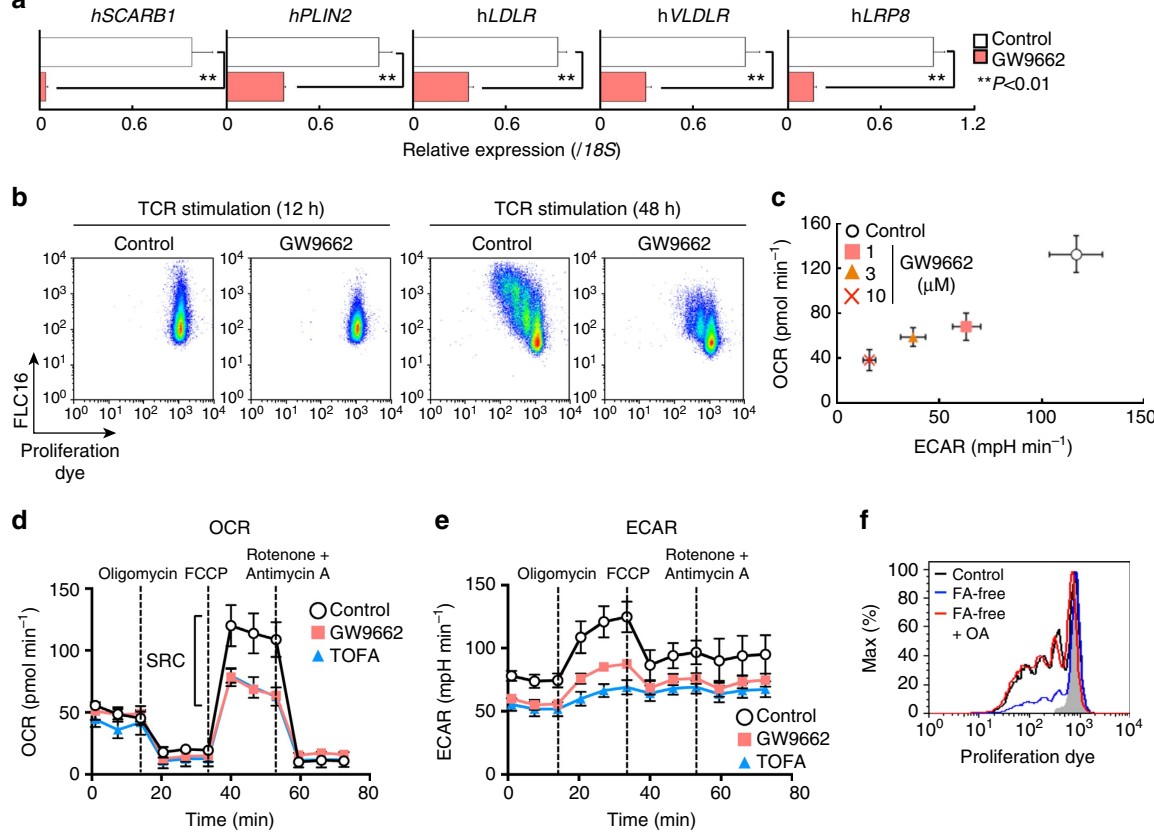

**Figure 7 | Anabolic FA metabolism is essential for activation of human CD4$^+$ T cells.** (**a**) qRT-PCR analyses of the relative expression of the genes encoding the enzymes and transporter in fatty acid uptake and lipolysis programmes in activated human CD4$^+$ T cells 24 h after TCR stimulation in the presence or absence of GW9662 (10 μM). (**P<0.01, Mann–Whitney U test) (**b**) Representative profiles of e670 and Bodipy FLC16 in activated human CD4$^+$ T cells in the presence or absence of GW9662. (**c**) Basal levels of OCR and ECAR of activated human CD4$^+$ T cells for 48 h with TCR stimulation in the presence or absence of indicated doses of GW9662. (**P<0.01, Mann–Whitney U test, n = 6 per group) (**d**) OCR of activated human CD4$^+$ T cells 48 h after TCR stimulation with or without GW9662 or TOFA under basal conditions and in response to sequential treatment with Oligomycin, FCCP and Rotenone-Antimycin A. (**e**) ECAR of activated human CD4$^+$ T cells 48 h after TCR stimulation with or without GW9662 or TOFA under basal conditions and in response to sequential treatment with Oligomycin, FCCP and Rotenone-Antimycin A as in **d**. (**f**) Representative plot of e670 proliferation dye in activated human CD4$^+$ T cells with or without oleic acid treatment under normal or fatty acid-free conditions 48 h after TCR stimulation was shown. Three technical replicates were performed for qRT-PCR (**a**). Six technical replicates were performed for Seahorse assay (**c–e**). The mean values with s.d. are shown (**a,c,d,e**). **P<0.01. Three independent experiments were performed with similar results (**a,b**). Two independent experiments were performed with similar results (**c–f**).

capacity for fatty acid uptake accompanied by decreased cell proliferation was detected (Supplementary Fig. 7a). Similarly, stimulation-induced proliferation was inhibited by the addition of TOFA (Supplementary Fig. 7d). The expression of genes associated with fatty acid uptake and lipolysis was reduced by GW9662 in activated human CD4$^+$ T cells (Fig. 7a). Consistent with these results, the capacity of fatty acid uptake and rapid proliferation after antigenic stimulation was also impaired by GW9662 treatment (Fig. 7b). Furthermore, we detected that control human CD4$^+$ T cells underwent metabolic reprogramming based on the level of OCR and ECAR after activation, and GW9662 treatment suppressed these events in a dose-dependent manner (Fig. 7c). We also measured OCR and ECAR following treatment with Oligomycin, FCCP and Rotenone with Antimycin A to again assess mitochondrial function, and GW9662- or TOFA-treated cells displayed a lower SRC and ECAR as was detected in activated murine CD4$^+$ T cells (Fig. 7d,e). Finally, supplementation of extrinsic oleic acid to the culture of activated human CD4$^+$ T cells under fatty acid-free conditions restored proliferation as was detected in stimulated mouse CD4$^+$ T cells (Fig. 7f). These results indicated that both fatty acid biosynthesis and fatty acid uptake programmes are

required for the early activation and proliferation of human CD4$^+$ T cells (Fig. 8).

## Discussion

Here we identified critical roles for mTORC1-mediated induction of PPARγ in the early activation of naive and memory CD4$^+$ T cells through the control of fatty acid uptake programmes. TCR–CD28 stimulation induces up-regulation of the genes associated with fatty acid uptake and fatty acid biosynthesis. Pharmacological inhibition or genetic silencing of PPARγ resulted in reduced expression of the genes associated with fatty acid uptake and impaired activation and rapid proliferation in both mice and human CD4$^+$ T cells. PPARγ directly bound to these genes and regulated their expression. Similarly, pharmacological inhibition of fatty acid biosynthesis suppressed TCR–CD28 stimulation-induced activation and proliferation of naive or memory CD4$^+$ T cells. Thus, our findings provide important mechanistic insights into fatty acid metabolism, particularly regarding the critical role for the fatty acid uptake programme that controls CD4$^+$ T cell activation and rapid proliferation.

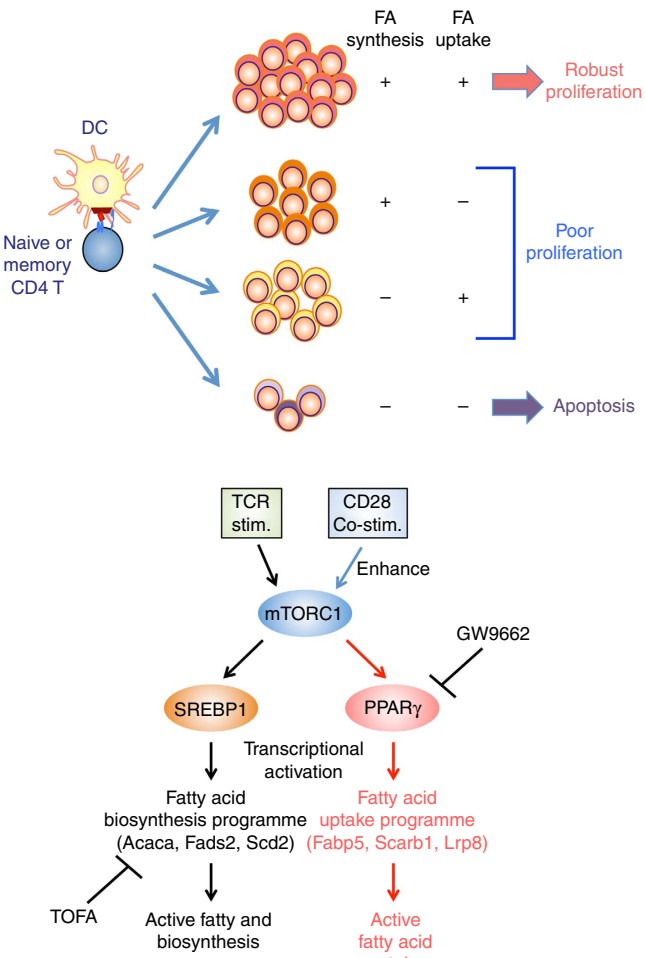

**Figure 8 | FA metabolism and early activation of CD4$^+$ T cells.** Top panel: antigenic stimulation induces both fatty acid biosynthesis and fatty acid uptake programmes in CD4$^+$ T cells. Both fatty acid biosynthesis and fatty acid uptake programmes are required for robust proliferation after antigenic stimulation. Inhibition of either fatty acid biosynthesis or fatty acid uptake programmes results in insufficient proliferation after antigenic stimulation. Activate CD4$^+$ T cells do not survive and undergo apoptosis when both fatty acid biosynthesis and fatty acid uptake programmes are dampen. Bottom panel: TCR/CD28-mTORC1 signalling axis controls fatty acid uptake and biosynthesis programmes through the induction of PPARγ and the activation of SREBP1 in activated CD4$^+$ T cells, respectively. PPARγ and SREBP1 directly bind to target genes and regulate their expression. GW9662: a selective, irreversible antagonist of PPARγ, TOFA: an allosteric inhibitor of ACC1.

The mechanisms that control fatty acid uptake and lipolysis in cell types other than adipocytes have not been well defined. Recently, resting quiescent memory CD8 T cells were reported to use cell-intrinsic lipolysis facilitated by extracellular glucose to support their metabolic demand rather than using extracellular fatty acids directly[31]. In contrast, our results indicate that activated naive and memory CD4$^+$ T cells actively acquire external fatty acids through the upregulation of mTOR-mediated PPARγ both *in vitro* and *in vivo* circumstances. T cells activated during graft-versus-host disease increased their fatty acid transport[41]. In addition, the uptake of fatty acid metabolites and their subsequent lipolysis were essential for the engagement of the activation of anabolic metabolism, prolonged survival and expression of key genes that mark commitment to M2 macrophage activation[33]. Furthermore, PPARγ is required for

orchestrating the metabolic programmes of M2 macrophage activation, including β-oxidation of fatty acids and mitochondrial biogenesis[42,43]. Thus, although intracellular fatty acids and lipolysis have essential roles in the activation, proliferation and function and the maintenance of immune cells, the mode of acquisition and/or its contribution in the regulation of the increased levels of fatty acids appeared to be distinct in different cell-types.

SREBPs were reported to be required for CD8 T cells to be fully activated[23] and Raptor-mTORC1-dependent metabolic reprogramming induced antigen-triggered exit from quiescence in T cells[44]. Kidani *et al.*[23] revealed that SREBP-induced cholesterols play a critical role in the activation and proliferation of CD8 T cells. In contrast, we demonstrate in this report that fatty acids are required for full activation and rapid proliferation of CD4$^+$ T cells. Consistent with the observation using *Raptor*-deficient mice by Yang and colleagues, we have identified that the mTOR-mediated signalling pathway is critical for SREBP1-induced fatty acid biosynthesis and rapid proliferation in both murine and human CD4$^+$ T cells. In addition to fatty acid biosynthesis, our study revealed that activated CD4$^+$ T cells utilize exogenous fatty acid for their nutrients through the up-regulation of PPARγ and its target genes associated with fatty acid uptake. Both pathways are required for the early and full activation and metabolic reprogramming of naive or memory CD4$^+$ T cells in murine and human systems. Consistent with our findings, hyperlipidemia caused by obesity or atherogenic conditions exhibited an increased frequency of CD44$^{hi}$ population among the CD4$^+$ T cells[40,45]. Highly proliferating cells including activated T cells appear to require large amounts of fatty acids. After TCR stimulation, primary CD4$^+$ T cells undergo at least five cell divisions during 5-day *in vitro* culture. As we demonstrated, the inhibition of either fatty acid biosynthesis or fatty acid uptake programmes dramatically blocked TCR stimulation-induced cell proliferation and cell growth. In addition, some tumour cells scavenge lipids or fatty acids from their environment in addition to the augmented fatty acid biosynthesis[46]. Furthermore, it is known that both fatty acid biosynthesis and fatty acid uptake programmes are required for adipocyte differentiation[47,48]. Importantly, PPARγ and SREBP1 directly bind and transactivate the expression of the genes involved in the fatty acid uptake and synthesis programmes. Although the precise mechanism of interrelationship between the fatty acid metabolic pathway and the glycolytic pathway remains to be completely defined, we detected reduced mTOR activity by the treatment of CD4$^+$ T cells with GW9662 (unpublished data). This could therefore be the reason why glycolysis is suppressed by the inhibition of fatty acid uptake in activated CD4$^+$ T cells.

Other important points revealed in this study are the requirement of fatty acids for the survival of CD4$^+$ T cells. If the fatty acid biosynthesis programme was inhibited by TOFA in activated CD4$^+$ T cells under fatty acid-free conditions, they did not survive well during *in vitro* culture. This was restored by the addition of OA. These observations indicate that fatty acid uptake programmes may substitute the fatty acid biosynthesis programmes and subsequent survival of activated T cells. It is unclear at present how fatty acids are metabolized in activated CD4$^+$ T cells. Lipidomics data showed that the level of triglycerides such as 1,2-dipalmitoyl-glycero-3-phosphoglycerol is increased by TCR stimulation in addition to the increase of phospholipids including phosphocholine, sphingosine and phosphoethanolamine. These results suggest that fatty acids, increased by changes in lipid metabolism after TCR stimulation, contribute to the cellular membrane in proliferating CD4$^+$ T cells. Furthermore, according to the results from

extrinsic supplementation of a series of fatty acids, the majority of saturated fatty acids and a portion of unsaturated fatty acids can rescue the proliferation of activated CD4$^+$ T cells. Especially, Myristic acid for saturated fatty acids, and Oleic acid for unsaturated fatty acids, almost completely restored proliferation of activated CD4$^+$ T cells cultured under fatty acid-free conditions. This is consistent with the Lipidomics data showing that 1-myristoyl-glycero-3-phosphocholine and 1-oleoyl-glycero-3-phosphocholine were the top-ranked metabolites increased by changes in lipid metabolites after TCR-stimulation. Therefore, these choline phosphoglycerides incorporated with fatty acids may play an essential role in the activation and rapid proliferation of CD4$^+$ T cells.

In conclusion, we demonstrate that in addition to *de novo* fatty acid biosynthesis, activated CD4$^+$ T cells utilize exogenous fatty acid through the TCR-mTOR-mediated up-regulation of expression of PPARγ and its downstream target genes. Further detailed studies focused on the molecular mechanisms underlying the fatty acid uptake and lipolysis programme may lead to a greater understanding of the role of fatty acid metabolism in immune cells such as T cells, and the discovery of potential therapeutic targets for the treatment of immune cell-mediated disorders.

## Methods

**Mice.** Transgenic mice used in this study were backcrossed to BALB/c or C57BL/6 mice 10 times. Anti-OVA-specific TCR-αβ (DO11.10) transgenic (Tg) mice were provided by Dr D. Loh (Washington University School of Medicine, St. Louis)[49]. All mice were housed under under SPF conditions and were used at 6–8 weeks of age. C57BL/6 mice and BALB/c mice were purchased from Clea Inc., Tokyo, Japan. All experiments using mice were approval by Chiba University Administrative Panel for Animal Care (# 28-181).

**Reagents.** The reagents used in this study were as follows: the FITC-, APC-, PE/Cy7, or BV421-conjugated anti-CD4 (GK1.5, 1 μg ml$^{-1}$), PE-conjugated anti-CD8 (53-6.7, 1 μg ml$^{-1}$), PE-, APC- and PE/Cy7-conjugated anti-CD62L (MEL-14, 1 μg ml$^{-1}$), FITC- and PE-conjugated anti-CD44 (IM7, 1 μg ml$^{-1}$), FITC or BV421-conjugated anti-human CD4 (A161A1, 1 μg ml$^{-1}$) and APC-conjugated anti-human CD45RO (UCHL1, 1 μg ml$^{-1}$) were purchased from BioLegend (San Diego, CA). The Alexa Fluor 488-conjugated anti-phospho ribosomal S6 protein (D57.2.2E, 0.5 μg ml$^{-1}$)was purchased from Cell signalling. GW9662 (Merck), TOFA (Merck) and rapamycin (Merck) were used as pharmacological inhibitors. Bodipy-labelled palmitate (Bodipy FLC16, 1 μM; Invitrogen), Bodipy-labelled LDL (Bodipy LDL, 10 μg ml$^{-1}$; Invitrogen) were used in conjunction with flow cytometry for uptake experiments. Intracellular neutral lipids were stained with 500 ng ml$^{-1}$ Bodipy 493/503 (Invitrogen) and fluorescence due to binding of Bodipy was measured by flow cytometry. Proliferation dye e670 were purchased from Invitrogen. Liquid Octanoic acid was adjusted to a final concentration of 100 mM and complexed to fatty acid-free BSA (Sigma). Other saturated (Decanoic acid, Dodecanoic acid, Myristic acid, Palmitic acid, Eicosanoic acid and Lignoceric acid) and unsatuated fatty acids (Palmitoleic acid, Linolenic acid, Linoleic acid, Oleic acid, Elaidic acid, Alachidonic acid, Docosahexanoic acid and Nervonic acid) were dissolved in DMSO to a final concentration of 100 mM and complexed to fatty acid-free BSA (Sigma). The OVA peptide (residues #323-339; ISQAVHAAHAEINEAGR) was synthesized by BEX Corporation, Tokyo, Japan. In some experiments, fatty acid-free BSA (Wako) was used for the culture medium.

**Mouse T cell cultures.** Naive (CD44$^{lo}$CD62L$^{hi}$) CD4$^+$ T cells were purified from the spleens of mice. After lysis of red blood cells, the CD4$^+$ T cells were obtained using a CD4$^+$ T cell isolation kit with anti-CD4 microbeads (Miltenyi Biotec), and naive CD44$^{lo}$CD62L$^{hi}$ cells were then further sorted to more than 99.5% purity using a FACS Aria cell sorter (BD Biosciences). Naive CD4 T cells were plated onto 24-well tissue culture plates (Costar) pre-coated with 10 μg ml$^{-1}$ agonistic anti-TCRβ antibody (H57-597) with 1 μg ml$^{-1}$ agonistic anti-CD28 antibody (clone 37.51, Biolegend). Non-polarized Th cell cultures contained IL-2 (15 ng ml$^{-1}$), anti-IL-4 antibody (BD Biosciences, BVD4-1D11) (1 μg ml$^{-1}$) and anti-IFNγ antibody (Biolegend, R4-6A2) (1 μg ml$^{-1}$). Oleic acid (Sigma-Aldrich) was dissolved in DMSO to a final concentration of 100 mM and was complexed to BSA. For lipid uptake experiments, naive CD4 T cells were cultured under indicated conditions in the presence of Bodipy FLC16, Bodipy LDL, or Bodipy 493/503, and mean fluorescence of Bodipy was measured by flow cytometry[31–33].

**Human T-cell cultures.** Whole blood was obtained from healthy donor volunteers with consent given (#1583). Human CD4$^+$ T cells were collected by a Ficoll

gradient. For human naive CD4$^+$ T cells, CD45RA$^+$CD45RO$^-$ cells were collected using a FACS Aria cell sorter (BD Biosciences). Human naive CD4$^+$ T cells were plated onto 48-well tissue culture plates (Costar) pre-coated with 1 μg ml$^{-1}$ anti-CD3 (clone OKT3) with 1 μg ml$^{-1}$ anti-CD28 (clone CD28.2) antibody. Non-polarized Th cell cultures contained IL-2 (15 ng ml$^{-1}$), anti-IL-4 antibody (1 μg ml$^{-1}$) and anti-IFNγ antibody (1 μg ml$^{-1}$).

**Metabolic profiling.** Unbiased metabolic profiling was performed by Human Metabolome Technologies, Inc. Primary naive CD44$^{lo}$CD62L$^{hi}$CD4$^+$ T cells were plated onto 24-well tissue culture plates (Costar) pre-coated with 10 μg ml$^{-1}$ agonistic anti-TCRβ antibody (H57-597) with 1 μg ml$^{-1}$ agonistic anti-CD28 antibody (clone 37.51, Biolegend) for non-polarized Th cell culture. At the end of culture (on day 2), fifty million cells were spun down and pellets were washed with 5% Mannitol before being frozen in methanol with internal standard solution (10 μM). All samples were analysed through LC–TOFMS at Human Metabolome Technologies, Inc.

**Immunoblotting assay.** Cytoplasmic extracts and nuclear extracts were prepared using the NE-PER Nuclear and Cytoplasmic Extraction Reagents (Thermo Fisher Scientific). The antibodies used for the immunoblot analyses were anti-SREBP1 (Abcam, 2A4) (2 μg ml$^{-1}$), anti-SREBP2 (Santa Cruz, H-164) (2 μg ml$^{-1}$) or (Abcam, synthesized peptide antigen) (2 μg ml$^{-1}$), anti-PPARγ (Abcam, synthesized peptide antigen) (1 μg ml$^{-1}$) and anti-Tubα (NeoMarkers, DM1A) (1 μg ml$^{-1}$). Full size images are presented in Supplementary Fig. 8.

**siRNA knockdown.** siRNA was introduced into activated non-polarized Th cells by electroporation using a human T-cell Nucleofector Kit and Nucleofector I (Amaxa). Activated Th cells were transfected with 675 pmol of control random siRNA or siRNA for *Srebf1*, *Srebf2* or *Pparg* (Applied Biosystems), and were cultured for 2 days under non-polarized conditions.

**Seahorse analysis.** The OCR and ECAR were measured with an XF96 analyzer (Seahorse Bioscience). Cultured CD4 T cells were seeded at a density of 200,000 cells per well of a XF96 cell culture microplate. Before assay, cells were equilibrated for 1 h in unbuffered XF assay medium supplemented with 25 mM glucose and 1 mM sodium pyruvate. Mixing, waiting and measure times were 2, 2 and 4 min, respectively. Compounds were injected during the assay at the following final concentrations: 0.2 μM Oligomycin, 0.5 μM FCCP and 0.75 μM Rotenone-Antimycin A.

**In vivo proliferation assay.** Naive CD4$^+$ T cells from OVA-specific DO11.10 Tg mice were transferred into wild-type BALB/c mice, and immunized with OVA/Alum. Immunized mice were injected intraperitoneally with DMSO or GW9662 at a dose of 5 mg kg$^{-1}$ per day. The treatment was started from same day with immunization and continued until the end of experiment. Mice were further intravenously injected with FLC16 2 h before euthanasia.

**The generation and culture of effector and memory Th2 cells.** Splenic CD62L$^+$CD44$^-$ naive KJ1$^+$CD4$^+$ T cells from DO11.10 OVA-specific TCR Tg mice were stimulated with an OVA peptide (Loh15, 1 μM) plus APC (irradiated splenocytes) under Th2-culture conditions for 6 days *in vitro*. Th2-condition; IL-2 (25 U ml$^{-1}$), IL-4 (10 U ml$^{-1}$) and anti-IFNγ mAb (1 μg ml$^{-1}$). These effector Th2 cells (3 × 10$^7$) were transferred intravenously into BALB/c *nu/nu* or BALB/c recipient mice. Five weeks after cell transfer, KJ1$^+$CD4 T cells in the spleen were purified by auto-MACS (Miltenyi Biotec) and cell sorting (BD Aria II) and then were used as memory Th2 cells.

**Quantitative real-time PCR.** Total RNA was isolated using the TRIzol reagent (Invitrogen). cDNA was synthesized using oligo (dT) primers and Superscript II RT (Invitrogen). Quantitative real time PCR was performed as described previously using an ABI PRISM 7500 Sequence Detection System. The primers and TaqMan probes were purchased from Applied Biosystems. The primers and Roche Universal probes used were purchased from Sigma and Roche, respectively. The gene expression was normalized using the *Hprt* mRNA signal or the *18S* ribosomal RNA signal.

**Chromatin immunoprecipitation assay.** ChIP assay was performed using ChIP assay kit (Millipore). In detail, 1 × 10$^6$ CD4 T cells were fixed with 1% paraformaldehyde at 37 °C for 10 min. Cells were sedimented, washed, lysed with SDS lysis buffer (50 mM Tris–HCl, 1% SDS, 10 mM EDTA, 1 mM phenylmethylsulfonyl fluoride, 1 μg ml$^{-1}$ aprotinin and 1 μg ml$^{-1}$ Leupeptin). The lysates were sonicated to reduce DNA lengths between 200 and 1,000 bp. The soluble fraction was diluted, precleared with salmon sperm DNA/protein A-agarose and then incubated with indicated antibodies. Then immune complexes were precipitated with protein A-agarose. The precipitated DNA was eluted with an elution buffer (0.1 M NaHCO$_3$ containing 1% SDS). The eluted material was

incubated at 65 °C for 4 h to reverse the formaldehyde cross-links, and DNA was extracted with phenol and chloroform. Ethanol-precipitated DNA was solubilized in water. The antibodies used in the ChIP assay were as follows: anti-trimethyl histone H3-K4 (Arbour assays, polyclonal) ($1 \mu g \, ml^{-1}$), anti-trimethyl histone H3-K27 (Millipore, 07-449) ($5 \mu g \, ml^{-1}$), anti-SREBP1 (Abcam, 2A4) ($5 \mu g \, ml^{-1}$), anti-PPARγ (Abcam) ($5 \mu g \, ml^{-1}$) and normal rabbit IgG Souther Biotech, 15H6) ($5 \mu g \, ml^{-1}$). Quantitative-PCR analyses were performed on an ABI prism 7500 real-time PCR machine with probes from the Roche Universal Probe Library System.

**Detection of apoptotic cells.** For the detection of apoptotic cells, Annexin V^FITC apoptosis detection kit II (BD Biosciences) was used according to the manufacturer's protocol.

**Statistical analysis.** Data are expressed as mean ± s.d. or mean ± s.e.m. Statistical analysis was performed with GraphPad Prism software programme (version 6). Differences were determined by using the two-tailed Student's $t$ test, or two-way ANOVA with Tukey's multiple comparison test. Differences with values of $P < 0.01$ or $P < 0.05$ were considered to be significant.

**Data availability.** The authors declare that the data supporting the findings of this study are available within the article and its Supplementary information files, or are available from the corresponding author upon request.

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

## Acknowledgements

We thank Kaoru Sugaya, Miki Kato, Toshihiro Ito and Yuka Masshardt for their excellent technical assistance. This work was supported by the Global COE Program (Global Center for Education and Research in Immune System Regulation and Treatment), and by grants from the Ministry of Education, Culture, Sports, Science and Technology (MEXT Japan) [Grants-in-Aid: for Scientific Research (S) #26221305, (B) #21390147 and 26293165, Young Scientists [A] #16H06224, and (B) #24790461,

Challenging Exploratory Research #26670362 and #23659240, Grant-in-Aid for Scientific Research on Innovative Areas #16H01352, and Scientific Research on Innovative Areas 'Stem Cell Aging' #26115009], AMED-CREST from Japan Agency for Medical Research and Development (AMED), the Ministry of Health, Labor and Welfare, The Astellas Foundation for Research on Metabolic Disorders, The Uehara Memorial Foundation, Osaka Foundation for Promotion of Fundamental Medical Research, Kanae Foundation for the Promotion of Medical Science, and Takeda Science Foundation.

## Author contributions

M.A., Y.E., D.J.T. and T.N. designed experiments, analysed the data and wrote the manuscript. M.A., Y.E., H.K.A. and T.Y. performed experiments. H.T. and K.Y. provided helpful suggestions during the manuscript preparation.

## Additional information

**Competing financial interests:** The authors declare no competing financial interests.

