## [Peer Review File · Nature Communications]

Reviewers' comments:

Reviewer #1

Expert in T-cell metabolism

(Remarks to the Author):

This is a well designed study examining the role of PPAR γ in regulating fatty acid metabolism in T cells. The data are robust and the work does a nice job of experimentally connecting fatty acid metabolism, mTOR and PPAR γ in T cells. That is the authors do a nice job in pulling together what has been known about fatty acid metabolism, mTOR and PPAR γ and relating it to T cells. There are a couple of points that the authors should try to clarify:

1. With regard to the presence and absence of CD28 signaling. Is it that without CD28 TCR signaling is suboptimal? Or is CD28 signaling providing a necessary distinct second signal. This could be tested by demonstrating maximal anti-TCR induced activation both in the presence and absence of anti-CD28.
2. Rapamycin experiments should demonstrate biochemical inhibition of mTORC1. Also, in general rapamycin is a mediocre inhibitor of T cell proliferation. At the doses that block metabolism, what is the effect on proliferation.
3. I don't exactly understand why treatment with GW inhibits glycolysis?
4. Why did the authors choose to use Th2 cells? This also begs the question, do the results also hold up for Th17 and Th1 cells?

Reviewer #2

Expert in mechanisms of metabolic disease and inflammation

(Remarks to the Author):

In the current study, Nakayama and colleagues demonstrate that PPAR γ dependent fatty acid uptake is critical for the metabolic reprogramming of T cells. This is a comprehensive and generally a well conducted series of experiments. It is significant because it highlights the importance of substrate metabolism in the etiology of clonal expansion of cells. I have some points that require clarification.

Major points.

1. PPAR γ dependent fatty acid uptake and OCR experiments. Although there were some experiments conducted using genetic silencing of PPAR γ , there is heavy bias on experiments utilizing GW9662, particularly in the Seahorse experiments. This is particularly worrying because Figure 3E shows a marked basal reduction in OCR prior to oligomycin treatment. It is reasonably well established that in vitro cell studies GW9662 displays considerable off-target effects (see Sargent et al *B J Pharmacol* 2004, Schaefer et al. *Int J Cancer* 2007). Therefore, these experiments should be repeated with siRNA.

2. Supplementation of FFA experiments. Why was only oleic acid (monounsaturated FFA) chosen for these experiments? Do long chain saturated fatty acids have the same effect given that these acids are very pro-inflammatory in immune cells?

Minor points.

1. There is a considerable literature on the role of PPAR γ in macrophage polarity (see work of Chawla et al, Olefsky et al). These important studies should be discussed.
2. Recent work from the Kallies laboratory has shown that metabolic reprogramming and clonal expansion of T cells is dependent on the transcription factor IRF4 (Man et al *Nat Immunol* 2013). Did IRF4 change in the model described here?
3. What is known about the role of hyperlipidemia and T cell activation in humans in vivo? This

should be discussed.

4. Why were non-parametric analyses used? One would assume that in cell culture experiments, the data would be normally distributed and, therefore, non parametric statistical analyses would appear to be inappropriate.

5. In each figure legend the authors described the number of biological replicates, but not the number of technical replicates. In addition in each figure legend the expression of data (eg mean \pm SEM) should be included.

Reviewer #3

Expert in FA metabolism
(Remarks to the Author):

This manuscript by Mulki et al. proposes that the mTORC1-PPAR γ pathway is crucial for the fatty acid (FA) uptake program in activated CD4 T cells and is required for full activation and proliferation of these cells. The general comments are that although showing that FA are needed for CD4 T cells activation might be novel, the manuscript presents a lot of data but the conclusions remain mostly correlative. The respective roles of FA biosynthesis, FA uptake and FA oxidation are muddled and unclear. Overall, the study relies too heavily on changes in gene expression that often are not adequately tested for functional significance.

Comments:

1 - The authors show changes in lipidomics and gene expression and interpret the data as an increase of the FA uptake program. It is not clear what the authors mean by "FA uptake" as it is often used to group genes of FA biosynthesis with those involving metabolism of exogenous FA.

2 - Data interpretation and follow up are often missing. Lipidomics are conducted and the altered lipid species are shown in the supplemental material but no interpretation of what the changes observed in the particular lipids might mean.

4- Expression of Ldlr and Vldlr is increased suggesting an increase in lipid uptake from lipoproteins as opposed to increased uptake of free exogenous FA. Why did the authors decide to test exogenous FA uptake and not lipid uptake from VLDL for example, which would seem to be more relevant in this case?

5 - The acylcarnitines are decreased which suggests reduced FA oxidation. If exogenous FA uptake is increased, and assuming FA oxidation is decreased, where is the FA that is taken up getting incorporated, into triglycerides? phospholipids etc? Also when exogenous FA uptake is increased why would cells synthesize more FA? Studies of FA metabolism using native FA and not bodipy FLC16 would be helpful.

5 - A large part of the data related to FA uptake is correlative. CD36 protein levels and membrane localization should be tested. Similarly, the role of FABP5 needs to be tested directly. Depleting CD36 or FABP5 would help in understanding the role of these proteins and consequently that of exogenous FA uptake in the activation/proliferation process. If CD36 protein expression is decreased in parallel with its mRNA, it could be consistent with the lipidomics-suggested reduction of FA oxidation since this protein regulates AMPK and oxidation of exogenous FA.

6 - Examining directly how the FA is metabolized (using native FA) and assessing FA biosynthesis from glucose would be much more informative than assaying uptake of Bodipy FLC16. It is not clear that Bodipy FLC16, which is slowly metabolized, is representative of "FA uptake". In addition few methodological details are included related to the accumulation of Bodipy FLC16 which make interpretation of what the data mean difficult.

- The authors propose that fatty acids are required for full activation and proliferation of CD4 T cells. This is based on experiments that inhibition of FA biosynthesis by TOFA coupled with FA deprivation impairs cell survival of activated CD4 T cells and that oleic acid addition restores survival. These observations are interesting but are not well developed. TOFA inhibits both FA biosynthesis and FA oxidation so the added FA could be providing the oxidative substrate. The role of the mitochondria and FA oxidation remains unclear and is barely commented on in the discussion.

Reviewer #1

Expert in T-cell metabolism

(Remarks to the Author):

This is a well designed study examining the role of PPAR γ in regulating fatty acid metabolism in T cells. The data are robust and the work does a nice job of experimentally connecting fatty acid metabolism, mTOR and PPAR γ in T cells. That is the authors do a nice job in pulling together what has been known about fatty acid metabolism, mTOR and PPAR γ and relating it to T cells. There are a couple of points that the authors should try to clarify:

1. With regard to the presence and absence of CD28 signaling. Is it that without CD28 TCR signaling is suboptimal? Or is CD28 signaling providing a necessary distinct second signal. This could be tested by demonstrating maximal anti-TCR induced activation both in the presence and absence of anti-CD28.

Response:

Thank you for the thoughtful and constructive review of our manuscript. According to the suggestion, we performed a set of experiments to examine the effect of increasing doses of TCR stimulation in the presence or absence of CD28-mediated signaling on the proliferation and fatty acid uptake by CD4 T cells. Increased fatty acid uptake and proliferation after antigenic stimulation was observed in the presence of CD28 co-stimulatory signals even under very strong TCR stimulation conditions (Supplementary Fig. 2). We think these data suggest that CD28 can provide a necessary additive signal for fatty acid uptake and rapid proliferation during antigenic stimulation. We included these results in the revised manuscript and revised the sentences in the Result section (page 9).

2. Rapamycin experiments should demonstrate biochemical inhibition of mTORC1. Also, in general rapamycin is a mediocre inhibitor of T cell proliferation. At the doses that block metabolism, what is the effect on proliferation.

Response:

According to this reviewer's suggestion, we checked the effect of rapamycin on T cell proliferation. As this reviewer thought, rapamycin treatment suppressed CD4 T cell proliferation at the doses that block fatty acid metabolism in mice and human systems (pages 10 and 20). We included these data in the revised manuscript.

3. I don't exactly understand why treatment with GW inhibits glycolysis?

Response:

We do not know exactly why the treatment with GW inhibits glycolysis in T cells, however we reproducibly detected reduced S6 kinase phosphorylation reflecting reduced mTOR activity after GW9662 treatment (Figure for Reviewer #1). This could therefore be the reason why glycolysis is inhibited by GW9662. We addressed this issue in the discussion section of the revised manuscript (pages 24-25).

4. Why did the authors choose to use Th2 cells? This also begs the question, do the results also hold up for Th17 and Th1 cells?

Response:

In response to the reviewer's suggestion, we assessed the role of fatty acid biosynthesis and fatty acid uptake in differentiating Th1 and Th17 cells. We first checked the expression of the genes associated with fatty acid biosynthesis and fatty acid uptake in differentiating Th1 and Th17 cells. Similar to Th2 and Th0 cells, the induction of the genes associated with fatty acid biosynthesis, fatty acid uptake, and lipolysis was detected in Th1 and Th17 cells. Consistent with these results, the proliferation of Th1 cells was substantially inhibited by TOFA treatment. Interestingly, however, the proliferation of Th17 cells was not apparently affected by TOFA treatment. Fatty acid uptake and proliferation after antigenic stimulation were both impaired by GW9662 treatment both in Th1 and Th17 cells. These data may suggest that Th17 cells mainly rely on fatty acid uptake, but not on fatty acid biosynthesis, in their proliferative capacity (Supplementary Fig. 4c). We included these results in the revised manuscript and revised the sentences in the Results section (page 14).

Reviewer #2

Expert in mechanisms of metabolic disease and inflammation

(Remarks to the Author):

In the current study, Nakayama and colleagues demonstrate that PPARgamma dependent fatty acid uptake is critical for the metabolic reprogramming of T cells. This is a comprehensive and generally well conducted series of experiments. It is significant because it highlights the importance of substrate metabolism in the etiology of clonal expansion of cells. I have some points that require clarification.

Major points.

1. *PPAR γ dependent fatty acid uptake and OCR experiments. Although there were some experiments conducted using genetic silencing of PPAR γ , there is heavy bias on experiments utilizing GW9662, particularly in the seahorse experiments. This is particularly worrying because Figure 3E shows a marked basal reduction in OCR prior to oligomycin treatment. It is reasonably well established that in vitro cell studies GW9662 displays considerable off-target effects (see Seargent et al B J Pharmacol 2004, Schaefer et al. Int J Cancer 2007). Therefore, these experiments should be repeated with siRNA.*

Response:

We thank this reviewer for the positive and thoughtful comments on our manuscript. As this reviewer suggested, we examined the effect of *Pparg* knockdown by siRNA in activated CD4 T cells in seahorse experiments. Similar to the effect of GW9662 experiments, the levels of OCR and ECAR were decreased in *Pparg*-knockdown CD4 T cells as compared to control CD4 T cells (Figures 4g and 4h). The basal reduction in OCR was also detected in *Pparg*-knockdown CD4 T cells. Therefore, we think the reduction was caused by the disruption of *Pparg* and not by off-target effects of GW9662. We included the results in the revised manuscript and revised the sentences in the Result section (page 15).

2. *Supplementation of FFA experiments. Why was only oleic acid (monounsaturated FFA) chosen for these experiments? Do long chain saturated fatty acids have the same effect given that these acids are very pro-inflammatory in immune cells?*

Response:

In response to this suggestion, we tested the effect of a series of fatty acids including saturated fatty acids, long chain saturated fatty acids and polyunsaturated fatty acids. While Dodecanoic acid and Lignoceric acid did not rescue the proliferation of activated CD4 T cells under fatty acid-free conditions, many of the saturated fatty acids restored the proliferation of activated CD4 T cells under fatty acid-free conditions (Figure 6g, upper). Especially, supplementation of Decanoic acid and Myristic acid in activated CD4 T cell cultures under fatty acid-free conditions almost completely restored the proliferation. In contrast, the majority of unsaturated fatty acids including polyunsaturated fatty acids other than Oleic acid and Elaidic acid did not rescue proliferation of activated CD4 T cells in fatty acid-free conditions (Figure 6g, lower). Alachidoic acid partially restored the proliferation of activated CD4 T cells in fatty acid-free conditions (Figure 6g, lower). Therefore, these results indicate that the majority of saturated fatty acids and some unsaturated fatty acids can rescue the proliferation of activated CD4 T cells. We included these results in Figure 6, and the revised sentences in the Results section (page 18).

Minor points.

1. *There is a considerable literature on the role of PPAR γ in macrophage polarity (see work of Chawla et al, Olefsky et al). These important studies should be discussed.*

Response:

In response to this suggestion, we described the role of PPAR γ in macrophage polarity in the discussion section in the revised manuscript (page 23).

2. *Recent work from the Kallies laboratory has shown that metabolic reprogramming and clonal expansion of T cells is dependent on the transcription factor IRF4 (Man et al Nat Immunol 2013). Did IRF4 change in the model described here?*

Response:

As this reviewer suggested, we assessed the expression of IRF4 in activated CD4 T cells in our models. mRNA and protein expression of IRF4 was not substantially changed by TOFA or GW9662 treatment. We included the results in Figures for reviewers #2.

3. *What is known about the role of hyperlipidemia and T cell activation in humans in vivo? This should be discussed.*

Response:

We added discussion of the current understanding of hyperlipidemia and T cell activation in humans *in vivo* in the discussion section (page 24).

4. *Why were non-parametric analyses used? One would assume that in cell culture experiments, the data would be normally distributed and, therefore, non parametric statistical analyses would appear to be inappropriate.*

Response:

As this reviewer suggested, we changed the statistic analysis from non-parametric to parametric analyses (see the Method and Figure legend sections).

5. *In each figure legend the authors described the number of biological replicates, but not the number of technical replicates. In addition in each figure legend the expression of data (eg mean \pm SEM) should be included.*

Response:

In response to this reviewer's suggestion, we included the number of technical replicates and specify what the error bars on each figure represent in each figure legend.

Reviewer #3

Expert in FA metabolism

(Remarks to the Author):

This manuscript by Mulki et al. proposes that the mTORC1-PPAR γ pathway is crucial for the fatty acid (FA) uptake program in activated CD4 T cells and is required for full activation and proliferation of these cells. The general comments are that although showing that FA are needed for CD4 T cells activation might be novel, the manuscript presents a lot of data but the conclusions remain mostly correlative. The respective roles of FA biosynthesis, FA uptake and FA oxidation are muddled and unclear. Overall, the study relies too heavily on changes in gene expression that often are not adequately tested for functional significance.

Comments:

1 - The authors show changes in lipidomics and gene expression and interpret the data as an increase of the FA uptake program. It is not clear what the authors mean by "FA uptake" as it is often used to group genes of FA biosynthesis with those involving metabolism of exogenous FA.

Response:

As this reviewer suggested, we more stringently used the terms "fatty acid biosynthesis", "fatty acid uptake" and "lipolysis" through the manuscript (see page 7, subhead, line 16 and line 21-22; page 9, line 14-15; page 10, line 7; page 11, subhead, and line 14; page 12, line 7; page 16, line 10, and page 20, line 11, line 18).

2 - Data interpretation and follow up are often missing. Lipidomics are conducted and the altered lipid species are shown in the supplemental material but no interpretation of what the changes observed in the particular lipids might mean.

Response:

According to this suggestion, we included possible interpretation of the changes detected in the particular lipid metabolites in the revised manuscript. For example, the level of triglycerides such as 1, 2-Dipalmitoyl-glycero-3-phosphoglycerol is increased by TCR stimulation in addition to the increase of phospholipids including phosphocholine, sphingosine, and phosphoethanolamine. These results suggest that fatty acids, increased by changes in lipid metabolism after TCR-stimulation, may contribute to the generation of plasma membrane in proliferating CD4 T cells. Furthermore, we tested the effect of a series of other fatty acids including long chain saturated fatty acids and polyunsaturated fatty acids on the rescue effect of CD4 T cell proliferation under fatty acid-free conditions (see also response to Reviewer #2, the second point). Among these, the majority of saturated fatty acids and some of the unsaturated fatty acids rescued the proliferation of activated CD4 T cells. Especially, Decanoic acid and Myristic acid (saturated fatty acids), and Oleic acid and Elaidic acid (unsaturated fatty acids), almost completely restored proliferative capacity of activated CD4 T cells cultured under fatty acid-free conditions. Lipidomics data showed that 1-Myristoyl-glycero-3-phosphocholine and 1-Oleoyl-glycero-3-phosphocholine were top-ranked metabolites which increased in lipid metabolites after TCR-stimulation. We included these points and more thoroughly discussed the potential roles of fatty acid metabolites in activated CD4 T cells in the discussion section of revised manuscript (pages 18, 19, 25 and 26).

4- Expression of Ldlr and Vldlr is increased suggesting an increase in lipid uptake from lipoproteins as opposed to increased uptake of free exogenous FA. Why did the authors decide to test exogenous FA uptake and not lipid uptake from VLDL for example, which would seem to be more relevant in this case?

Response:

In response to this comment, we assessed LDL or nonpolar neutral lipids uptake using LDL complex or neutral lipids conjugated with fluorescent dye (Supplementary Fig. 4a). Although the uptake of LDL or neutral lipid was induced by TCR stimulation, much less effect of GW9662 was observed in the reduction of uptake of these lipid metabolites than fatty acid uptake. Furthermore, we tested the effect of cholesterol supplementation on the cell survival and proliferation in activated CD4 T cells under fatty acid-free conditions (Supplementary Fig. 6a and 6b). However, extrinsic supplementation of cholesterol did not restore the proliferation or cell survival of activated CD4 T cells. These results are included in Supplementary Fig. 4 and 6 in the revised manuscript (pages 13, 14, 18 and 19).

5 - The acylcarnitines are decreased which suggests reduced FA oxidation. If exogenous FA uptake is increased, and assuming FA oxidation is decreased, where is the FA that is taken up getting incorporated, into triglycerides? phospholipids etc? Also when exogenous FA uptake is increased why would cells synthesize more FA? Studies of FA metabolism using native FA and not bodipy FLC16 would be helpful.

Response:

We think fatty acids that are taken up are incorporated into both triglycerides and phospholipids in activated CD4 T cells. This is because the two metabolites (1, 2-Dipalmitoyl-glycerol-3-phosphoglycerol and 1-Myristoyl-glycerol-3-phosphocholine) are the top-ranked metabolites induced by TCR stimulation as listed in supplementary Figure 1. With regard to the second point raised by this reviewer, highly proliferating cells, including activated T cells, may require very large amounts of fatty acids. After TCR-stimulation, primary CD4 T cells undergo at least 5 cell divisions during 5-day *in vitro* cultures. As we demonstrated, the inhibition of either fatty acid biosynthesis or fatty acid uptake programs dramatically blocked the TCR stimulation-induced cell proliferation and cell growth (Figures 1f, 4b, 4c, 5f, 5h, and 7b). In addition, some tumor cells scavenge lipids or fatty acids from their environment in addition to the augmentation of fatty acid biosynthesis (Currie et al., *Cell Metabolism*, 18, 153-, 2013). Furthermore, it is known that both fatty acid biosynthesis and fatty acid uptake programs are required for adipocyte differentiation (Rodriguez et al., *Am J Physiol Endocrinol Metab*, 309, E691-, 2015 and Liu et al., *Acta Pharmacol Sin*, 8, 1052-, 2004). We mentioned and discussed these issues in the discussion section (pages 24, 25 and 26).

It is indeed very interesting to trace how fatty acids taken up are metabolized in activated CD4 T cells. However, this is a large and complex issue to investigate, and therefore we want to study and address this issue in a more comprehensive way and report the results in a separate paper in the near future (see also our response to the 6th comment of this reviewer).

5 - A large part of the data related to FA uptake is correlative. CD36 protein levels and membrane localization should be tested. Similarly, the role of FABP5 needs to be tested directly. Depleting CD36 or FABP5 would help in understanding the role of these proteins and consequently that of exogenous FA uptake in the activation/proliferation process. If CD36 protein expression is decreased in parallel with its mRNA, it could be consistent with the lipidomics-suggested reduction of FA oxidation since this protein regulates AMPK and oxidation of exogenous FA.

Response:

As this reviewer suggested, we checked the CD36 protein expression in activated CD4 T cells. The expression of cell surface CD36 was unchanged or rather decreased in activated cells as compared to naïve CD4 T cells in parallel with mRNA expression (Supplementary Fig. 1b). Thus, it could be consistent with the lipidomics data as this reviewer suggested. We included these data in Supplementary Fig. 1b, and describe the results in the revised manuscript (page 8). With regard to the role of FABP5 in fatty acid uptake, we examined the effect of *Fabp5* knockdown in the proliferation and fatty acid uptake in activated CD4 T cells (Figures for Reviewers #3). Fatty acid uptake and proliferation after antigenic stimulation was impaired by the reduced expression of *Fabp5*.

6 - Examining directly how the FA is metabolized (using native FA) and assessing FA biosynthesis from glucose would be much more informative than assaying uptake of Bodipy FLC16. It is not clear that Bodipy FLC16, which is slowly metabolized, is representative of "FA uptake". In addition few methodological details are included related to the accumulation of Bodipy FLC16 which make interpretation of what the data mean difficult.

- The authors propose that fatty acids are required for full activation and proliferation of CD4 T cells. This is based on experiments that inhibition of FA biosynthesis by TOFA coupled with FA deprivation impairs cell survival of activated CD4 T cells and that oleic acid addition restores survival. These observations are interesting but are not well developed. TOFA inhibits both FA biosynthesis and FA oxidation so the added FA could be providing the oxidative substrate. The role of the mitochondria and FA oxidation remains unclear and is barely commented on the discussion.

Response:

It would be indeed very interesting to trace how fatty acids taken up are metabolized and assessing FA biosynthesis using native FA in activated CD4 T cells. However, this is a large issue that has not been well analyzed in primary cells for multiple fatty acids at the same time, and we are of the view that several sets of complex and carefully designed experiments would need to be performed to address this question sufficiently well for publication. In addition, (i) it is currently difficult to analyze the metabolic fates of a number of fatty acids at the same time, and (ii) we think we have a robust data set presented within this manuscript (a point made by Reviewer #1) and believe that the data presented supports the conclusions made within the manuscript. Therefore, we want to study and address this issue in a more comprehensive way and report the results in a separate focused study in the near future. In response to the reviewer's comment, we included clear methodological information on the Bodipy FLC16 experiments with accompanying references. In fact, many investigators have very recently use Bodipy FLC16 to measure the uptake of FA (O' Sullivan et al., *Immunity*, 41, 1-, 2014, Everts et al., *Nat. Immunol*, 15, 323-, 2014, and Huang et al., *Nat. Immunol*, 9, 846-, 2014) (page 8, line 7).

In response to the reviewer's comment regarding the role of the mitochondria and fatty acid oxidation, we analyzed the expression of *cpt1*, which mediates the first step in long-chain fatty acid import into mitochondria, in activated CD4 T cells with or without TOFA treatment. The expression of *Cpt1a* and *Cpt1b* in activated CD4 T cells was not changed by TOFA or GW9662 treatment (Supplementary Fig. 6c). The expression of *Cpt1c* was slightly decreased by the treatment of activated CD4 T cells with TOFA or GW9662 (Supplementary Fig. 6c). The levels of mitochondrial membrane potential were not substantially affected by TOFA or GW9662 treatment (Supplementary Fig. 6d). Furthermore, we also examined the effect of fatty acid supplementation to TOFA-treated activated CD4⁺ T cells on the mitochondrial respiratory capacity (Supplementary Fig. 6e and 6f). Extrinsic supplementation of Oleic acid to the TOFA-treated activated CD4⁺ T cell cultures under fatty acid-free conditions clearly restored the levels of SRC and ECAR (Supplementary Fig. 6e and 6f). These results suggest that extrinsically supplemented fatty acids could be providing

the oxidative substrate in mitochondria as pointed out by this reviewer. We included these results in the revised manuscript and revised the sentences in the Result section (page 19).

Legends of Figure for Reviewers

Figure for Reviewers #1

Intracellular staining profiles of p-S6 protein and Bodipy FLC16 in stimulated CD4⁺ T cells for 48 hours after TCR stimulation with or without GW9662.

Figure for Reviewers #2

(a) qRT-PCR analyses of *Irf4* in naïve CD4 T cells, and activated CD4⁺ T cells with or without TOFA, or GW9662. (b) Intracellular staining profiles of IRF4 in CD4⁺ T cells for 24 hours after TCR stimulation with or without TOFA, or GW9662.

Figure for Reviewers #3

(a) Representative profiles of e670 in stimulated CD4⁺ T cells with or without transduction of siRNA for *Fabp5*. (b) Representative profiles of Bodipy FLC16 in stimulated CD4⁺ T cells with or without transduction of siRNA for *Fabp5*.

Figure for Reviewers #1

Figure for Reviewers #2

Figure for Reviewers #3

REVIEWERS' COMMENTS:

Reviewer #1 (Remarks to the Author):

The authors have addressed all of my concerns

Reviewer #2 (Remarks to the Author):

The authors have done an excellent job addressing my concerns.

Reviewer #3 (Remarks to the Author):

The authors have done a great job in addressing previously raised concerns. A few comments:

1- It would help the reader and enhance impact of the paper to provide a simple diagram summarizing the major steps involved in mTOR/PPAR γ induction of fatty acid metabolic reprogramming of T cells. This could be included as a panel in figure 6 or 7.

2- Fig. 6: The new data showing effects of different fatty acids clutter the figure and could be condensed to show only the fatty acids that were effective. Fatty acids tested and found inefficient could be mentioned in the figure legends and or text.

3- Also Fig 6 and text line 356: What is Alachidoic acid? Is this supposed to be arachidonic acid?

Reviewer #3 (Remarks to the Author):

The authors have done a great job in addressing previously raised concerns. A few comments:

1- It would help the reader and enhance impact of the paper to provide a simple diagram summarizing the major steps involved in mTOR/PPAR γ induction of fatty acid metabolic reprogramming of T cells. This could be included as a panel in figure 6 or 7.

Response:

In response to reviewer's comment, we included a schematic diagram summarizing the major steps involved in mTOR/PPAR γ induction of fatty acid metabolic reprogramming of T cells in Figure 8.

2- Fig. 6: The new data showing effects of different fatty acids clutter the figure and could be condensed to show only the fatty acids that were effective. Fatty acids tested and found inefficient could be mentioned in the figure legends and or text.

Response:

According to this reviewer's suggestion, we removed some of the data of fatty acids supplementation experiment that shows inefficient restoration.

3- Also Fig 6 and text line 356: What is Alachidoic acid? Is this supposed to be arachidonic acid?

Response:

According to the suggestion, we corrected Alacidoic acid to Arachidonic acid.